# GROUP-LABEL-FREE VALIDATION FOR SPURIOUSLY CORRELATED DATA

## ABSTRACT

Deep learning models are known to be sensitive to spurious correlations between irrelevant features and the labels. These spurious features can negatively affect the model's generalization and robustness, particularly for groups consisting of examples without spurious correlations. Early approaches address this issue by requiring group labels in the training set, and more recent methods aim to reduce reliance on group labels in training; however, many state-of-the-art approaches still require a validation set with group annotations for hyperparameter tuning or model selection, which are often unavailable or costly to obtain. In this work, we propose SIEVE, a plug-and-play module that constructs a group-aware validation set for robust model evaluation under spurious correlations, without using any group annotations. SIEVE identifies confusing training examples based on feature-space similarity, and iteratively separates them into spurious and non-spurious subsets based on differing loss dynamics patterns, which we discovered in our data analysis. The selected samples are assigned pseudo group labels and used as a surrogate validation set for model selection. Our method is annotation-efficient, easy to implement, and compatible with existing methods that rely on group-labeled validation sets for hyperparameter tuning and model selection. Experiments on benchmark datasets demonstrate that SIEVE enables robust model selection without access to group labels, achieving performance competitive with methods that use true group annotations.

## 1 INTRODUCTION

Deep neural networks often exploit spurious correlations between non-causal features (or shortcuts) that frequently co-occur with the labels in the training data, leading to severely degraded performance on minority or atypical data (Shah et al., 2020). For example, an image classifier may rely on whether the background is water or land to classify whether a bird is a water bird or a land bird, causing it to fail when the background does not match the bird type (Sagawa et al., 2020). While spurious features enable models to achieve high average accuracy, they cause the models to consistently fail on non-spuriously correlated groups (e.g., "land bird-on-water" and "water bird-on-land"), which are typically under-represented in the dataset.

Early works generally tackled spurious correlations by manually dividing the training data into different predefined groups, e.g., according to the class label and whether spurious correlations are present. A common strategy for minimizing the worst-group loss is to use a loss that reweighs the groups, such as distributionally robust optimization (Sagawa et al., 2020; Oren et al., 2019), maximum weighted loss discrepancy (Khani et al., 2019). Another approach is to learn a model that is invariant to group differences, such as learning an invariant representation across groups as in IRM (Arjovsky et al., 2019), or by employing learned data augmentations combined with consistency regularization to enforce classifier invariance to group differences as in CAMEL (Goel et al., 2021). In practice, identifying and labeling all minority groups is prohibitively expensive and sometimes infeasible. For example, in chest X-ray datasets (Wang et al., 2017), labels may be spuriously correlated with hospital-specific artifacts such as scanner tags, stripes, or medical devices that co-occur with pneumonia (Zech et al., 2018). Assigning group labels (such as whether spurious features exist) to each image in such scenarios would require extensive domain expertise and manual effort.

Recent works have designed group-label-efficient methods that require few group labels (Kirichenko et al., 2023; Liu et al., 2021; Nam et al., 2022; Zhang et al., 2022), or group-label-free methods that require no group labels (Nam et al., 2020; Taghanaki et al., 2022; Zhu et al., 2023; Wu et al., 2023; Zheng et al., 2024). Group-label-efficient methods such as JTT (Liu et al., 2021), DFR (Kirichenko et al., 2023), AFR (Qiu et al., 2023) and DaC (Noohdani et al., 2024), improve worst-group accuracy without group labels on training data. JTT (Liu et al., 2021) first trains a standard model to identify misclassified examples, which is hypothesized to belong to minority groups. Then it upweights them in a second training stage. DFR (Kirichenko et al., 2023) first trains a standard ERM model, then retrains the last layer on a group-balanced validation set with known group labels. AFR (Qiu et al., 2023) adopts the same last-layer retraining paradigm but replaces the group-balanced set with a loss-weighted held-out split, so group labels are used only for hyperparameter tuning and model selection on the validation set. DaC (Noohdani et al., 2024) similarly improves group robustness via last-layer retraining, likewise using group labels only for validation-time hyperparameter tuning and model selection. These approaches significantly narrow the performance gap between group-label-efficient and group-supervised training. However, they still rely on group-labeled validation sets for model selection or hyperparameter tuning, and each method designs its own strategy. This lack of a unified, general-purpose selection mechanism limits their practicality. In parallel, several group-label-free approaches has been proposed to improve robustness without access to any group labels (Zhu et al., 2023; Taghanaki et al., 2022; Wu et al., 2023; Zheng et al., 2024). These methods offer diverse strategies, such as leveraging slow learning rates or analyzing saliency maps, which are often architecture-specific, and require some additional knowledge (e.g., the use of a captioning model in LBC (Zheng et al., 2024)). While these methods are effective, they cannot be easily integrated into existing training pipelines.

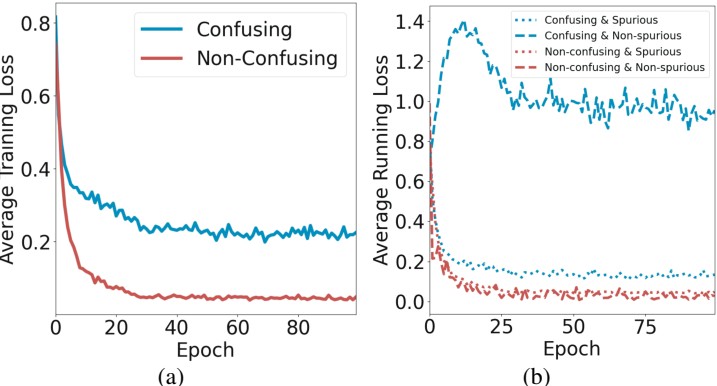

(a)  (b)

Figure 1: (a) Training losses of 100 confusing and 100 non-confusing examples from the Waterbirds dataset. (b) The same examples in (a), further split by spurious and non-spurious groups.

Our work contributes a novel algorithm, **SIEVE** (*Spurious-aware IterativE Validation Example selection*), for constructing pseudo group labels, without relying on the architectures used or additional knowledge. SIEVE is motivated by our observation of differing loss dynamics of the spurious examples and non-spurious examples in the set of *confusing* training examples, where a confusing example is one which is close to the other classes. Unsurprisingly, confusing examples have higher training losses than non-confusing ones as they are harder to classify, as illustrated in Figure 1(a). What is interesting is that if we further split the confusing examples into spurious and non-spurious examples, then the training loss of spurious examples decreases rapidly, while that of non-spurious examples decreases more slowly or even increases as illustrated by Figure 1(b) and Figure 2. This suggests that the model is likely using the spurious features as shortcuts to help minimize loss when they are present, even at the expense of increasing the loss of examples without spurious correlations. Such a difference between the loss dynamics of spurious and non-spurious examples is not evident in the set of non-confusing examples, likely because the examples are far away from the decision boundary and thus easy to classify. We provide a more detailed explanation of our observation on the loss dynamics in Section 2.1. Additional analyses of the loss dynamics on other datasets are presented in Section C.1. Using the above observation, SIEVE offers a general method to construct

a group-labeled validation set by selecting potentially spurious and non-spurious examples from the confusing category based on the loss dynamics of the examples.

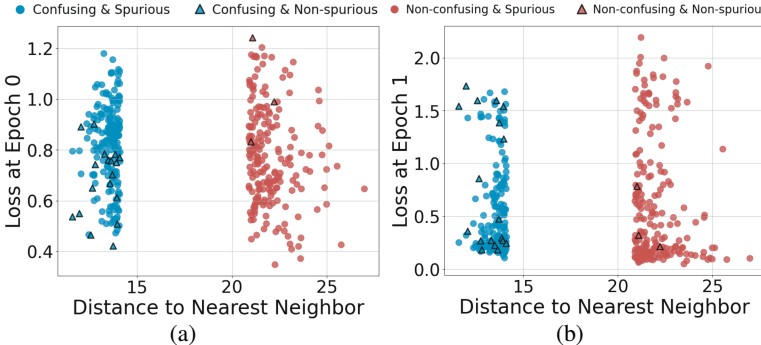

Figure 2: (a) The training loss of examples in four subgroups of Waterbirds on epoch 0. (b) The training loss of examples in four subgroups of Waterbirds on epoch 1. Here, the x-axis denotes the distance to the nearest other-class example. We take the 200 examples with the least distance as the confusing examples, and the 200 examples with the largest distance as the non-confusing ones, as done in Eq. (2) and Eq. (3) respectively, resulting in two sharp boundaries defined by two threshold distances values.

SIEVE provides a drop-in replacement of ground-truth group-labeled validation set for model selection and evaluation in the presence of spurious correlations. It can thus be plugged into existing group-label-efficient methods for handling spurious correlations, without modifying them. We empirically demonstrate the strong performance of SIEVE by integrating SIEVE for validation for standard ERM and state-of-the-art methods like JTT, AFR and DaC: the models selected by the SIEVE validation set perform competitively with those selected by a validation set with ground-truth group labels. Due to space constraints, a more comprehensive review of related works can be found in Section A.

## 2 METHOD

We provide a conceptual illustration to further explain the loss dynamics pattern discussed in the introduction, describe our SIEVE algorithm, and show how the pseudo group-labeled set it produces can be plugged into existing group-label-efficient training pipelines.

### 2.1 LOSS DYNAMICS

We provide a conceptual illustration in Figure 3 to explain how the loss dynamics pattern observed in the introduction can arise. Section D.1.4 provides a simulation study to more formally investigate when and why this arises.

The figure illustrates a binary classification dataset, with a true decision boundary ("Decision Boundary") determined by the causal features (the vertical axis). On the other hand, the high correlation of the spurious features with the labels allows them to provide a high-accuracy spurious decision boundary ("Wrong Boundary"). The confusing examples are those that lie close to the true decision boundary (yellow and orange regions), while the spurious examples are those that can be correctly separated by the spurious decision boundary (orange and grey regions).

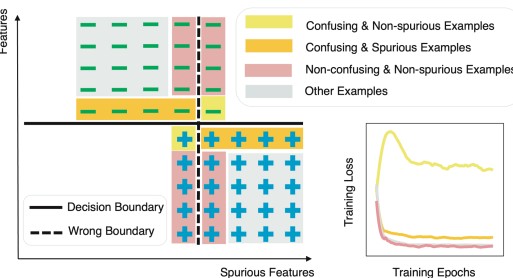

Figure 3: Conceptual illustration of how the loss dynamics pattern in the introduction may arise.

---

**Algorithm 1** SIEVE

---

**Input**: Training dataset $D_{\text{tr}}$, feature extractor $f_\theta^{\text{feat}}$, number of iterations $T_{\text{iter}}$
**Output**: Selected validation set $V$
 1: Initialize $V \leftarrow \emptyset$
 2: Extract feature representations: $\mathbf{Z} \leftarrow f_\theta^{\text{feat}}(\mathbf{X})$
 3: Compute distance: $d_i \leftarrow \min_{y_j \neq y_i} \text{dist}(\mathbf{z}_i, \mathbf{z}_j)$
 4: **for** $t = 1$ to $T_{\text{iter}}$ **do**
 5:     Compute confusing examples $S_{\text{c}}$ and non-confusing examples $S_{\text{nc}}$ using Eqs. (2) and (3)
 6:     Initialize model $f_{\theta_t}$ and compute its training loss $L^{(0)}$ on $D_{\text{tr}}$
 7:     Train $f_{\theta_t}$ on $D_{\text{tr}}$ for one epoch, then recompute its training loss $L^{(1)}$ on $D_{\text{tr}}$
 8:     Compute loss change: $\Delta L \leftarrow L^{(1)} - L^{(0)}$
 9:     Compute spurious examples $S_{\text{s}}$ and non-spurious examples $S_{\text{ns}}$ using Eqs. (5) and (6)
10:     **for all** $(\mathbf{x_i}, y_i) \in S_{\text{s}} \cup S_{\text{ns}}$ **do**
11:         Assign $\hat{s}_i \leftarrow 0$ if $i \in S_{\text{s}}$, else $\hat{s}_i \leftarrow 1$
12:         Assign $\hat{g}_i$ based on $y_i$ and $\hat{s}_i$
13:     **end for**
14:     Update $V$ to $V \cup \{(\mathbf{x_i}, y_i, \hat{g}_i)\}_{i \in S_{\text{s}} \cup S_{\text{ns}}}$, and $D_{\text{tr}}$ to $D_{\text{tr}} \setminus (S_{\text{s}} \cup S_{\text{ns}})$
15: **end for**
16: **return** $V$

---

While the true decision boundary appears to be easy to learn in the figure, in reality, this is much harder to learn as compared to the spurious decision boundary.

When training starts for a randomly initialized model, the strong correlation between the spurious features and the output is often exploited by the learning algorithm. The average loss for the confusing spurious examples will quickly decrease as the spurious decision boundary can correctly classify them. On the other hand, for the confusing non-spurious examples, the average loss will show less improvement, because the spurious decision boundary will struggle with correctly classifying them, and may even perform worse than a random initial decision boundary.

For the non-confusing examples, which lie far away from the true decision boundary, the classification task is relatively easy. As a result, many of them will be correctly classified by at least one of the two decision boundaries, and in some cases by both, regardless of whether the examples are spurious or non-spurious. In particular, as illustrated in the figure, the spurious decision boundary may classify some non-spurious examples correctly, but there are still non-confusing non-spurious examples that it misclassifies.

## 2.2 SIEVE

The pseudocode of SIEVE is shown in Algorithm 1. At a high-level, SIEVE iteratively identifies confusing samples via feature-space similarity, and then classify their group membership based on their loss dynamics.

We provide a detailed explanation for each step below. Let the training dataset be $D_{\text{tr}} = \{(\mathbf{x}_i, y_i)\}_{i=1}^{n_{\text{tr}}} \subset \mathbb{R}^d \times \{-1, 1\}$, where each $\mathbf{x}_i \in \mathbb{R}^d$ is an input feature and $y_i \in \{-1, 1\}$ is its label. We use $\mathbf{X}$ to denote all the feature vectors $(\mathbf{x_1}, \mathbf{x_2}, \ldots, \mathbf{x_{n_{\text{tr}}}})$ and $\mathbf{Y}$ all the labels $(y_1, y_2, \ldots, y_{n_{\text{tr}}})$.

SIEVE first computes the feature vectors and distances between them, which will remain fixed when iteratively selecting validation examples. We extract feature vectors $\mathbf{z}_{1:n} = \mathbf{Z} = f_\theta^{\text{feat}}(\mathbf{X})$ using a pretrained model $f_\theta^{\text{feat}}$. Our main experiments use a ResNet-50 pretrained on ImageNet (He et al., 2016). A pretrained model is also required in other works on spurious correlations, typically as the initial model that will be finetuned to optimize the worst-case group performance. Then, for each sample $\mathbf{x}_i$, we compute the minimum distance to any other example from the opposite class in the feature space:

$$d_i = \min_{j : y_j \neq y_i} \text{dist}(\mathbf{z}_i, \mathbf{z}_j). \tag{1}$$

While computing all pairwise distances can be expensive for very large datasets, this can be done efficiently on datasets with tens of thousands of examples particularly when using a GPU. For a

very large dataset, we can approximate the minimum distances by randomly sampling a subset of examples (i.e., $\mathbf{z}_j$ is selected from this subset instead from the entire dataset in Eq. (1)).

During the iterative selection phase, each iteration consists of four main steps. We first select the top $N_c$ samples with the smallest $d_i$ as *confusing examples*:

$$S_c = \{(\mathbf{x_i}, y_i) \mid i \in \mathrm{argmin}_{N_c}\{d_i \mid i \in D_{\mathrm{tr}}\}\}, \tag{2}$$

and the top $N_{nc}$ samples with the largest $d_i$ as *non-confusing examples*:

$$S_{nc} = \{(\mathbf{x_i}, y_i) \mid i \in \mathrm{argmax}_{N_{nc}}\{d_i \mid i \in D_{\mathrm{tr}}\}\} \tag{3}$$

We then estimate the loss changes for each sample as follows: initialize a new model $f_{\theta_t}$ and compute the initial training losses $L^{(0)} = \mathcal{L}(f_{\theta_t}(\mathbf{X}), \mathbf{Y})$. Now train the model for one epoch on the dataset $D_{\mathrm{tr}}$, and compute the updated losses $L^{(1)} = \mathcal{L}(f_{\theta_t}(\mathbf{X}), \mathbf{Y})$. The loss changes for the examples are

$$\Delta L = L^{(1)} - L^{(0)}. \tag{4}$$

With the loss changes computed above, we select the top $K$ samples with the most negative $\Delta L_i$ (i.e., largest loss decrease) from the confusing examples as *spurious samples*:

$$S_s = \{(\mathbf{x_i}, y_i) \mid i \in \mathrm{argmin}_K\{\Delta L_i \mid i \in S_c\}\}, \tag{5}$$

and the top $K$ samples with the most positive $\Delta L_i$ (i.e., largest loss increase) from the confusing examples as non-spurious samples:

$$S_{ns} = \{(\mathbf{x_i}, y_i) \mid i \in \mathrm{argmax}_K\{\Delta L_i \mid i \in S_c\}\} \tag{6}$$

Note that all selected examples come from the confusing set and a detailed discussion is provided in Section C.3. Each selected sample is assigned a binary spurious category label $\hat{s}_i$, which is 0 if $i \in S_s$, and 1 if $i \in S_{ns}$. We also assign a group label $\hat{g}_i$ based on the original label $y_i$ and the spurious label $\hat{s}_i$.

Finally, the identified spurious and non-spurious examples are removed from the training set and added to the validation set: $D_{\mathrm{tr}} \leftarrow D_{\mathrm{tr}} \setminus (S_s \cup S_{ns})$,

$$V \leftarrow V \cup \{(\mathbf{x_i}, y_i, \hat{g}_i)\}_{i \in S_s \cup S_{ns}}.$$

This process repeats for $T_{\mathrm{iter}}$ rounds, and the final $V$ is the final validation set. Although selection is performed iteratively, the additional computational overhead is modest in practice; we report the running time of the selection process in Section C.4.

## 2.3 INTEGRATING SIEVE FOR VALIDATION

After constructing a validation set with pseudo group labels, we use it to perform model selection and, optionally, hyperparameter tuning, without requiring access to true group annotations. The SIEVE-constructed validation set can be plugged into any training pipeline that relies on group-labeled validation data for robust evaluation.

We illustrate this with standard empirical risk minimization (ERM), as shown in Algorithm 2. Given a SIEVE-constructed validation set $V$, we train a new model $f_{\theta}^{\mathrm{ERM}}$ on a refined dataset $D_{\mathrm{tr}'}$, which is the original training set with the validation samples excluded $D_{\mathrm{tr}'} = D_{\mathrm{tr}} \setminus V$. The per-sample training loss is defined as: $L^{\mathrm{ERM}} = \mathcal{L}(f_{\theta}^{\mathrm{ERM}}(\mathbf{X}'), \mathbf{Y}')$, where $\mathbf{X}'$ and $\mathbf{Y}'$ represent the input features and labels of the refined dataset $D_{\mathrm{tr}'}$.

We then evaluate the model's performance on $V$ by computing the worst-group accuracy: $A_{\mathrm{wg}} = \min_{\hat{g}} \; \mathrm{Accuracy}(f_{\theta}^{\mathrm{ERM}}(\mathbf{X}_{\hat{g}}), \mathbf{Y}_{\hat{g}})$, where $\hat{g}$ is the pseudo group label assigned by SIEVE. The best model checkpoint is selected based on the highest $A_{\mathrm{wg}}$ observed during training: $f_{\theta*}^{\mathrm{ERM}} = \arg\max_{\theta} A_{\mathrm{wg}}(\theta)$.

Although we use ERM as a simple demonstration, SIEVE is compatible with a wide range of robust training algorithms, such as JTT, AFR and DaC, which typically require group-labeled validation sets. Since SIEVE is independent of the training procedure itself, it serves as a drop-in replacement for group-labeled validation sets, making it broadly applicable and easy to integrate into existing pipelines. In our experiments, we show that replacing the validation set in these methods with the one constructed by SIEVE leads to strong performance, even without using any true group annotations.

---

**Algorithm 2** ERM Training with Model Selection Using the SIEVE-Constructed Validation Set

---

**Input:** Updated $D_{\text{tr}'} = \{(\mathbf{x_i}, y_i, \hat{g}_i)\}_{i=1}^{n_{\text{tr}'}}$, validation set $V = \{(\mathbf{x_i}, y_i, \hat{g}_i)\}_{i \in S_V}$, loss function $\mathcal{L}$, total epochs $T$

**Output:** Best model parameters $\theta^*$

1: Initialize model parameters $\theta$, and set $A_{\text{best}}$ to 0 and $\theta^*$ to $\theta$.
2: **for** $t = 1$ to $T$ **do**
3:     Train $f_\theta^{\text{ERM}}$ on $D_{\text{tr}'}$ using ERM
4:     Compute worst-group accuracy: $A_{\text{wg}} = \min_{\hat{g}} \text{Accuracy}(f_\theta^{\text{ERM}}(\mathbf{X}_{\hat{g}}), \mathbf{Y}_{\hat{g}})$
5:     If $A_{\text{wg}} > A_{\text{best}}$, then update $A_{\text{best}}$ to $A_{\text{wg}}$ and $\theta^*$ to $\theta$,
6: **end for**
7: **return** $\theta^*$

---

## 3 EXPERIMENTS

We first demonstrate that integrating SIEVE into existing methods yields comparable or even slightly better performance compared to using the original group-labeled validation set for hyperparameter tuning and model selection, across four vision benchmarks and one NLP benchmark.

**Datasets.** We use five datasets known for exhibiting significant worst-group performance drops under standard ERM training due to spurious correlations. Descriptions of these datasets are presented below and more details about the datasets are in Section B.1.1.

- Waterbirds (Sagawa et al., 2020): Created by combining bird photos from Caltech-UCSD Birds-200-2011 data (Wah et al., 2011) with image backgrounds from the Places dataset (Zhou et al., 2018). Birds are labelled as waterbirds or landbirds, with backgrounds of water or land, where waterbirds are typically on water backgrounds and landbirds on land. The target is bird type, and the spurious feature is background type.

- Dominoes (Pagliardini et al., 2023): Combines images of CIFAR10 vehicles with MNIST digits, as done in Noohdani et al. (2024). Each CIFAR10 vehicle (automobile or truck) is placed over an MNIST digit (0 or 1). The target is the vehicle type, and the spurious feature is the digit.

- Metashift (Liang & Zou, 2022): The set up follows Wu et al. (2023). The target is to classify cats and dogs, while the spurious feature is the objects and backgrounds.Backgrounds present in the val set and test set (shelf) are not used in the training set.

- CelebA (Liu et al., 2015): Contains images of celebrity faces. The target is hair color, while the spurious feature is gender, as proposed by Sagawa et al. (2020).

- MultiNLI (Williams et al., 2018): Contains pairs of sentences for natural language inference. The target is the logical relationship (entailment, neutral, or contradiction), while the spurious feature is the presence of negation words in the second sentence. This setup follows Sagawa et al. (2020), who identify that due to annotation artifacts, negation words are significantly correlated with the contradiction label.

**Baselines.** We consider four baselines: ERM and three state-of-the-art group-label efficient methods, namely, JTT (Liu et al., 2021), AFR (Qiu et al., 2023) and DaC (Noohdani et al., 2024). ERM trains a standard model on the training set with a group-labeled validation set for model selection. The other three methods use a group-labeled validation set for tuning the method-specific hyperparameters and model selection. More details about the descriptions of the methods and the method-specific hyperparameters are described in Section B.1.2 and Section B.1.3.

### 3.1 COMPARISON WITH BASELINES

Table 1 compares SIEVE-enhanced versions of ERM, JTT, AFR and DaC to their original counterparts that rely on group-labeled validation sets. We evaluate worst-group and average accuracy across five benchmarks, comprising four vision datasets (Waterbirds, Metashift, Dominoes, CelebA) and one non-vision dataset (MultiNLI), and find that SIEVE provides competitive performance on both modalities.

Table 1: A comparison of mean and worst-group accuracy on the test sets of baseline methods, including methods applying SIEVE, on five datasets. The results are in the form of the mean (std) for every method. The bolded values indicate the best results.

| Method | Waterbirds | | Metashift | | Dominoes | | CelebA | | MultiNLI | |
|---|---|---|---|---|---|---|---|---|---|---|
| | Worst | Average | Worst | Average | Worst | Average | Worst | Average | Worst | Average |
| ERM | 77.83 (1.60) | 86.18 (0.56) | 66.18 (3.67) | 75.53 (1.14) | 77.31 (3.76) | 88.40 (1.13) | 72.59 (4.63) | 92.41 (0.21) | 68.15 (1.71) | 81.30 (0.18) |
| ERM-SIEVE | 79.84 (2.07) | 93.74 (0.46) | 65.73 (2.39) | 76.26 (1.67) | 84.69 (2.92) | 91.73 (0.93) | 72.59 (1.79) | 92.91 (0.14) | 68.44 (2.52) | 82.04 (0.21) |
| JTT | 87.88 (0.26) | 90.12 (0.64) | 67.36 (4.71) | 75.88 (1.69) | 78.16 (2.0) | 88.68 (0.85) | 80.13 (1.65) | 89.60 (3.32) | 72.50 (1.85) | 79.10 (0.30) |
| JTT-SIEVE | 78.52 (1.51) | 93.52 (0.32) | 71.62 (3.43) | 77.66 (1.13) | 84.76 (1.85) | 91.92 (0.75) | 79.07 (1.72) | 86.93 (0.35) | 72.16 (1.34) | 79.28 (0.06) |
| AFR | 88.03 (5.08) | 92.47 (1.76) | 71.67 (1.00) | 72.57 (2.83) | 81.00 (3.32) | 90.40 (2.42) | 82.97 (2.63) | 95.20 (0.10) | **73.90 (0.46)** | 81.67 (0.15) |
| AFR-SIEVE | 88.10 (1.25) | 93.17 (0.87) | 71.57 (0.61) | 72.60 (2.60) | 83.73 (3.35) | 91.23 (0.84) | **82.97 (0.86)** | 95.20 (0.10) | 73.21 (1.07) | 81.34 (0.49) |
| DaC | **92.30 (0.40)** | 95.30 (0.40) | 78.30 (1.60) | 79.30 (0.10) | 89.20 (0.10) | 92.20 (0.30) | 79.08 (1.05) | 88.73 (3.56) | – Not Applicable – | |
| DaC-SIEVE | 91.71 (1.95) | 94.06 (1.29) | **81.37 (0.60)** | 82.87 (0.24) | **89.59 (0.69)** | 92.11 (0.56) | 78.61 (0.40) | 88.90 (0.09) | | |

Table 2: Group-wise and overall (micro-averaged) precision (%) of pseudo group labels assigned by SIEVE on seed 1 across different datasets. Values in parentheses indicate the number of correctly assigned samples over the total number in that group.

| Dataset | Spurious Groups | | Non-Spurious Groups | | Micro-averaged Precision | Descriptions |
|---|---|---|---|---|---|---|
| | y = 0 | y = 1 | y = 0 | y = 1 | | |
| Waterbirds | 92.51 (568/614) | 96.12 (99/103) | 90.00 (18/20) | 45.71 (32/70) | 88.85 (717/807) | y=0: landbird |
| Metashift | 94.96 (132/139) | 86.67 (39/45) | 33.33 (1/3) | 64.29 (9/14) | 90.05 (181/201) | y=0: cat |
| Dominoes | 88.34 (379/429) | 84.05 (680/809) | 77.55 (38/49) | 81.82 (45/55) | 78.99 (218/276) | y=0: digit 0 |
| CelebA | 97.47 (77/79) | 4.76 (1/21) | 1.95 (4/205) | 99.21 (501/505) | 71.98 (583/810) | y=0: non-blonde |

**ERM vs. ERM-SIEVE.** ERM-SIEVE achieves improved average accuracy across all datasets compared to standard ERM. Notably, on Dominoes, ERM-SIEVE improves worst-group accuracy by more than 7 points (from $77.31\%$ to $84.69\%$). On Waterbirds, we observe a 2 points improvement in worst-group accuracy. On Metashift, CelebA, and MultiNLI, ERM-SIEVE maintains comparable worst-group performance while slightly improving average accuracy. These results show that even a standard ERM pipeline benefits from using SIEVE for model selection.

**JTT vs. JTT-SIEVE.** On Dominoes and Metashift, JTT-SIEVE achieves higher worst-group and average accuracy than the original JTT. Specifically, worst-group accuracy improves by over 6 points on Dominoes and by more than 4 points on Metashift, highlighting that SIEVE provides a more reliable signal for identifying and upweighting minority samples. On Waterbirds, while JTT achieves higher worst-group accuracy, JTT-SIEVE produces substantially higher average accuracy. On CelebA, JTT-SIEVE exhibits a slight trade-off with marginally lower worst-group accuracy but maintains competitive performance without requiring group annotations. On MultiNLI, JTT-SIEVE achieves performance comparable to JTT in terms of both worst-group and average accuracy.

**AFR vs. AFR-SIEVE.** AFR-SIEVE preserves AFR's strong performance. Notably, on Dominoes, we observe a clear improvement in worst-group accuracy from $81.00\%$ to $83.73\%$. Across the other datasets, AFR-SIEVE performs comparably to AFR. On CelebA, AFR-SIEVE achieves the highest worst-group accuracy ($82.97\%$) and average accuracy ($95.20\%$) among all methods.

**DaC vs. DaC-SIEVE.** SIEVE improves DaC's worst-group performance on Metashift and Dominoes. In particular, on Metashift, DaC-SIEVE improves worst-group accuracy by over 3 points ($78.3\%$ to $81.37\%$) and increases average accuracy from $79.3\%$ to $82.87\%$. On Dominoes, DaC-SIEVE slightly outperforms DaC on worst-group accuracy and achieves similar average accuracy. While DaC with ground-truth validation labels achieves the highest worst-group accuracy on Waterbirds ($92.3\%$), DaC-SIEVE achieves competitive performance ($91.71\%$) without requiring any group annotations. On CelebA, DaC-SIEVE demonstrates robust performance with comparable worst-group accuracy and slightly improved average accuracy.

**Summary.** Across all methods and datasets, SIEVE consistently shows competitive performance in both worst-group and average accuracy, despite not using any group labels. This demonstrates that the SIEVE-validation set serves as an effective replacement for ground-truth group-labeled validation sets across both vision and language tasks.

Table 3: Comparison of different feature extractors for SIEVE. While the downstream model (ERM or DaC) remains the same, we vary the architecture used during SIEVE selection (either ResNet-50 or VGG16). All VGG16 results are averaged over 3 random seeds, while ResNet-50 results are averaged over 5 seeds (as reported in Table 1).

| Method | SIEVE Arch | Waterbirds | | Metashift | | Dominoes | |
|--------|------------|------------|------|-----------|------|----------|------|
| | | Worst | Avg | Worst | Avg | Worst | Avg |
| ERM-SIEVE | ResNet-50 | 79.84 (2.07) | 93.74 (0.46) | 65.73 (2.39) | 76.26 (1.67) | 84.69 (2.92) | 91.73 (0.93) |
| ERM-SIEVE | VGG16 | 79.23 (3.34) | 94.06 (0.48) | 57.44 (1.89) | 72.61 (0.92) | 79.23 (3.34) | 94.06 (0.48) |
| DaC-SIEVE | ResNet-50 | 91.71 (1.95) | 94.06 (1.29) | 81.37 (0.60) | 82.87 (0.24) | 89.59 (0.69) | 92.11 (0.56) |
| DaC-SIEVE | VGG16 | 89.04 (2.13) | 94.78 (0.30) | 77.26 (6.42) | 81.45 (2.63) | 89.66 (1.05) | 91.97 (0.55) |

### 3.2 THE ACCURACY OF THE PSEUDO GROUP LABELS

We evaluate the quality of pseudo group labels assigned by SIEVE on seed 1 across three datasets, since the results are consistent across different random seeds. Table 2 reports group-wise and overall labeling precision, along with the raw counts of correct predictions. In Waterbirds, Metashift and Dominoes, Spurious groups constitute the majority. As shown, SIEVE generally achieves a high accuracy of at least 85% on these majority groups, indicating its reliability in identifying easy-to-separate examples.

By contrast, CelebA presents a different scenario. The dataset is dominated by female examples, which spuriously correlate with blonde hair. This imbalance persists across both target classes, leading to high precision in groups containing blonde females (spurious, y=0) and non-blonde females (non-spurious, y=1). However, precision drops significantly in the two male groups, indicating difficulty in identifying minority group members when they are underrepresented and less separable.

Across all datasets, minority groups which have extremely limited sample sizes are generally harder to label accurately. For instance, the cat images in the non-spurious group of Metashift account for only 3.1% of the dataset (37 out of 1,204 examples), and the landbird images in the non-spurious group of Waterbirds comprise just 3.2% of all training samples (189 out of 5,994 examples). Nonetheless, the overall labeling accuracy remains high (e.g., 88.85% on Waterbirds), and the constructed validation set is still effective for guiding model selection and tuning. Even when group labels are incorrect, the selected samples still lie in regions of the feature space that are confusing to the model—typically exhibiting rapid increases in training loss. These confusing examples continue to provide useful signals for identifying robust model checkpoints.

That said, methods that rely heavily on accurate identification of the minority group (such as JTT, which upweights presumed minority samples) may be more sensitive to such labeling noise. For instance, in Waterbirds, the relatively low precision on the landbird images in the non-spurious group may explain the reduced effectiveness of JTT when integrated with SIEVE. 

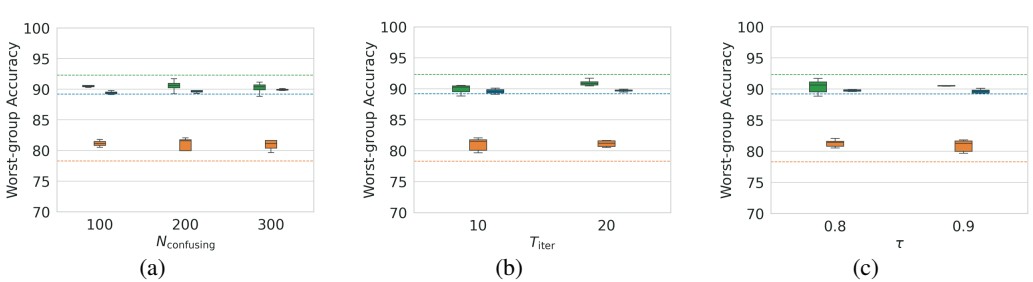

Figure 4: Sensitivity of DaC-SIEVE to (a) number of confusing examples $N_c$, (b) selection iterations $T_{iter}$, and (c) threshold $\tau$. Each boxplot shows worst-group accuracy across datasets, based on Table 9 with $T_{iter}=5$ excluded due to instability from insufficient validation selection. Dashed lines indicate DaC baselines. DaC-SIEVE remains robust across hyperparameter settings.

## 3.3 SENSITIVITY ANALYSIS FOR SIEVE

To evaluate the robustness of SIEVE under different settings, we conduct two sets of sensitivity analyses: one examining the architecture used during SIEVE selection, and the other exploring the effects of hyperparameter choices. We find that SIEVE remains robust across architectures and hyperparameter configurations. Below, we provide detailed analyses to support this finding.

**Effect of architecture used for SIEVE selection.**   In our main experiments, we use a ResNet-50 model pretrained on ImageNet for the SIEVE stage. To test whether SIEVE remains effective when using a different model architecture, we replace the ResNet-50 with VGG16 (Simonyan & Zisserman, 2014), also pretrained on ImageNet, during the selection process. Note that the downstream training (e.g., ERM or DaC) still uses ResNet-50; only the model used to generate training dynamics and select validation examples is changed. Results are shown in Table 3. We observe that SIEVE remains effective even when the model architecture for selection is switched to VGG16. For instance, ERM-SIEVE with VGG16 achieves similar worst-group accuracy as with ResNet-50 on Waterbirds and Dominoes. On Metashift, although the performance drops slightly, the results are still competitive. These findings suggest that SIEVE is not tightly coupled to any specific model architecture and can generalize across different feature extractors. Importantly, we directly reused the hyperparameters from ResNet-50 (e.g., learning rate and weight decay) without tuning for VGG16. This highlights that SIEVE is not tightly coupled to a specific model and remains robust across different architectures and configurations.

**Effect of hyperparameters.**   We further evaluate SIEVE sensitivity to hyperparameters beyond the main configurations reported in Section 3.1, where we set $\tau = 0.8$, $N_c = 200$ and $T_{\text{iter}} = 20$ for Waterbirds and Dominoes; $N_c = 100$ and $T_{\text{iter}} = 10$ for Metashift. We perform a grid search over $N_c \in \{100, 200, 300\}$ and $T_{\text{iter}} \in \{5, 10, 20\}$, under two confidence thresholds $\tau \in \{0.8, 0.9\}$ for DaC-SIEVE. The results on Waterbirds, Metashift and Dominoes, averaged over three random seeds, are summarized in Figure 4, with detailed results provided in Table 9 in Section C.2. The bolded numbers in the table indicate the main experimental results reported earlier.

We find that DaC-SIEVE achieves stable performance across a wide range of hyperparameter configurations, with worst-group accuracy varying only modestly under different settings. This indicates that SIEVE's selection process is relatively robust to changes in $\tau$, $N_c$, and $T_{\text{iter}}$.

A clear trend emerges with respect to the number of selection iterations $T_{\text{iter}}$: $T_{\text{iter}} = 5$ generally leads to worse performance than 10 or 20 iterations. For example, in Waterbirds (with $\tau = 0.8$ and $N_c = 200$), increasing $T_{\text{iter}}$ from 5 to 20 improves worst-group accuracy from 88.68% to 91.71%. Thus a small number of iterations may be insufficient to identify diverse or high-quality confusing examples.

In contrast, varying the number of confusing examples $N_c$ between 100 and 300 has less consistent impact. Performance remains relatively stable across this range, especially when $T_{\text{iter}}$ is sufficiently large. This indicates that SIEVE can tolerate a fairly wide range of candidate pool sizes without significant performance degradation.

As for the confidence threshold $\tau$, both $\tau = 0.8$ and $\tau = 0.9$ lead to strong and comparable results. While there is no consistently dominant value, we observe that $\tau = 0.8$ offers slightly more stable worst-group accuracy across seeds in several configurations.

All three hyperparameters influence the number and quality of examples selected into the validation set. Our results suggest that as long as these values are set within a reasonable range that allows SIEVE to collect a sufficiently large and diverse set of confusing examples, the method performs consistently well. In particular, very small values of $T_{\text{iter}}$ or overly strict thresholds $\tau$ can hurt performance because they limit the selection too much, especially when $N_c$ is also small. This overall stability is further illustrated in Figure 4, which summarizes worst-group accuracy across different combinations of $T_{\text{iter}}$, $N_c$, and $\tau$ (excluding $T_{\text{iter}} = 5$). The boxplots show that once a reasonable number of iterations is used, DaC-SIEVE remains stable and effective across all datasets.

## 4 CONCLUSION

We proposed SIEVE, a plug-and-play framework for constructing pseudo group-labeled validation sets in the presence of spurious correlations, without requiring any group annotations. SIEVE identifies confusing samples via feature similarity and groups them using training loss dynamics. The resulting validation set enables group-aware model selection and hyperparameter tuning in a fully group-label-free setting. Through experiments on five benchmark datasets, we demonstrated that models using SIEVE for validation selection achieve comparable or even better worst-group and average performance than those relying on true group labels. SIEVE can be seamlessly integrated into existing robust learning methods such as JTT and DaC, offering a practical and effective alternative to expensive or unavailable group annotations. By bridging the gap between group-label-efficient methods and fully group-label-free model selection, SIEVE paves the way toward more scalable and robust deep learning under spurious correlations.

## 5 ETHICS STATEMENT

We acknowledge the ICLR Code of Ethics. This paper submission does not raise questions regarding the Code of Ethics.

## 6 REPRODUCIBILITY STATEMENT

We will release the codes of our algorithm to reproduce the experimental results. Experiments use public datasets, and all the detailed experimental settings, including random seeds and hyperparameters, are reported in the paper in detail.

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

## A  RELATED WORK

This section is organized by decreasing reliance on group supervision: we first review methods that assume full group labels during training, then those that train label-free but still tune on a group-labeled validation set, and finally approaches that dispense with group labels altogether, setting the stage for our SIEVE framework, which removes the remaining need for any group annotations.

## A.1 Group Robustness Methods Requiring Group Labels

Deep neural networks often exploit simplicity bias – they latch onto the simplest, most easily learned features even if those are spurious correlations, while ignoring more complex but relevant features (Shah et al., 2020). This bias causes models to reply on spurious attributes that happen to correlate with the label in the training set.

As a result, models achieve high average accuracy but can fail on minority groups where these spurious correlations do not hold. Early methods (Khani et al., 2019; Oren et al., 2019; Sagawa et al., 2020) to combat the group robustness challenge assume group labels (identifying which samples contain the spurious attribute) are available for each example. A prime example is Group-DRO (Sagawa et al., 2020), which optimizes the worst-case loss across the predefined groups. By leveraging true group annotations, GroupDRO significantly improves worst-group accuracy on benchmarks.

However, these methods assume full knowledge of the spurious attribute or group for each example, which may not hold in real-world scenarios.

## A.2 Group-label-Efficient Approaches

To reduce reliance on costly group labels, a variety of approaches improves worst-group performance without the group labels for the training set, but just the validation set. For example, JTT (Liu et al., 2021) first trains a standard ERM model and records the misclassified training examples, then retrains a second model with these misclassified examples upweighted. A validation set with group labels is used to tune the number of epochs in the first stage and the upweighting factor in the second. SSA (Nam et al., 2022) uses a small set of group-labeled samples to train a group attribute estimator and assign pseudo-group labels to the rest of the training data. CnC (Zhang et al., 2022) begins by training an ERM model to identify same-class samples with dissimilar spurious features. It subsequently employs contrastive learning to align the representations of these samples. Hyperparameter tuning is performed using a validation set with group annotations. DFR (Kirichenko et al., 2023) first trains a standard ERM model, and then retrains the last layer using a group-balanced validation set with known group labels. AFR (Qiu et al., 2023) retrains the last layer of a standard ERM-trained model using a weighted loss that emphasizes examples where the base model performs poorly, thereby automatically upweighting the minority group. A validation set is used for hyperparameter tuning and model selection. DaC (Noohdani et al., 2024) is also a method that improves the group robustness by last-layer retraining. It made a further move on reducing the reliance on group labels, as it only uses group labels of the validation set for hyperparameter tuning and model selection.

These methods demonstrate that it is possible to approach the worst-group performance of Group DRO without the group labels of the training set. But they still lean on group annotations for hyperparameter tuning or model selection. This motivates us to propose SIEVE, a plug-in framework that can be integrated into existing group-label-efficient methods to eliminate their reliance on group-labeled validation sets. By doing so, SIEVE fills a key gap in current approaches and pushes group-robust training towards more practical, fully group-label-free scenarios.

## A.3 Group Robustness without Group Labels

In the absence of known group labels, recent work has explored ways to infer or approximate them. For instance, SSL (Zhu et al., 2023) observed that samples conflicting with spurious correlations are learned at a slower rate. Leveraging this insight, they introduced a speed-aware sampling strategy to encourage the model to prioritize these slow-learning samples. MaskTune (Taghanaki et al., 2022) assumes that ERM-trained models predominantly attend to image regions that exhibit strong spurious correlations with the label. Based on this assumption, it identifies and masks regions with the highest xGradCAM scores, and subsequently fine-tunes a new model on the masked data. DISC (Wu et al., 2023) constructs a bank of human-interpretable concepts, identifies those whose relevance varies across environments as spurious, and mitigates spurious correlations by intervening on the training data. LBC (Zheng et al., 2024) mitigates spurious correlations by using a vision-language model to detect attributes, measure their spuriousness, and guide relabeling and balanced training based on this information.

Overall, these approaches offer diverse strategies to approximate or replace group supervision, highlighting the importance of developing group-label-free methods for addressing spurious correlations. Building on this insight, our work proposes a complementary direction that identifies group-robust learning signals directly from training dynamics, without relying on any group annotations or pretrained concept knowledge as in existing group-label-free methods.

# B   TECHNICAL DETAILS AND ADDITIONAL EXPERIMENTS

In the following parts, we provide additional implementation details, extended analyses, and additional experiments to support the findings in the main paper.

In Section B.1 of this section, we provide the details about the datasets, hyperparameter settings, and hardware configurations used in our experiments.

In Section C, we provide further analyses for SIEVE, including:

- Validation of the core observation on loss dynamics across four datasets (Waterbirds, Dominoes, Metashift and CelebA), demonstrating its generality (Section C.1).

- Complete hyperparameter sensitivity analysis results showing the robustness of SIEVE to hyperparameter configurations (Section C.2).

- A discussion about selecting validation data only from the confusing set, with supporting statistics and visualizations (Section C.3).

- A runtime comparison to assess the computational overhead introduced by SIEVE (Section C.4).

- A discussion about the limitations of SIEVE (Section C.5).

## B.1   ADDITIONAL EXPERIMENT DETAILS

### B.1.1   DATASETS

We summarize the details about the datasets we use in Tables 4 to 7.

| | **Target**: bird type; **Spurious feature**: background type. | | | |
|---|---|---|---|---|
| **Group** $g$: | 0 | 1 | 2 | 3 |
| **Target** $y \in \{0, 1\}$: | 0 (landbird) | 0 (landbird) | 1 (waterbird) | 1 (waterbird) |
| **Spurious** $s$: | 0 (land) | 1 (water) | 0 (land) | 1 (water) |
| **# Train data:** | 3,498 (73%) | 184 (4%) | 56 (1%) | 1,057 (22%) |
| **# Val data:** | 467 | 466 | 133 | 133 |
| **# Test data:** | 2,255 | 2,255 | 642 | 642 |

Table 4: Waterbirds Dataset.

| | **Target**: vehicle type; **Spurious feature**: digit. | | | |
|---|---|---|---|---|
| **Group** $g$: | 0 | 1 | 2 | 3 |
| **Target** $y \in \{0, 1\}$: | 0 (automobile) | 0 (automobile) | 1 (truck) | 1 (truck) |
| **Spurious** $s$: | 0 (digit 0) | 1 (digit 1) | 0 (digit 0) | 1 (digit 1) |
| **# Train data:** | 3,592 (45%) | 398 (5%) | 398 (5%) | 3,592 (45%) |
| **# Val data:** | 500 | 499 | 499 | 500 |
| **# Test data:** | 490 | 490 | 490 | 490 |

Table 5: Dominoes Dataset.

### B.1.2   BASELINES

- ERM trains a standard model on the training set with a group-labeled validation set for model selection.

| **Target**: animal category; **Spurious feature**: background type. | | | | |
| --- | --- | --- | --- | --- |
| **Group $g$:** | 0 | 1 | 2 | 3 |
| **Target $y \in \{0, 1\}$:** | 0 (cat) | 0 (cat) | 1 (dog) | 1 (dog) |
| **Spurious $s$:** | 0 (sofa / bed) | 1 (shelf) | 0 (bench / bike) | 1 (shelf) |
| **# Train data:** | 611 | 0 | 512 | 0 |
| **# Val data:** | 0 | 37 | 0 | 44 |
| **# Test data:** | 0 | 198 | 0 | 262 |

Table 6: Metashift Dataset.

| **Target**: hair colour; **Spurious feature**: gender. | | | | |
| --- | --- | --- | --- | --- |
| **Group $g$:** | 0 | 1 | 2 | 3 |
| **Target $y \in \{0, 1\}$:** | 0 (non-blonde) | 0 (non-blonde) | 1 (blonde) | 1 (blonde) |
| **Spurious $s$:** | 0 (female) | 1 (male) | 0 (female) | 1 (male) |
| **# Train data:** | 71,629 (44%) | 66,874 (41%) | 22,880 (14%) | 1,387 (1%) |
| **# Val data:** | 8,535 | 8,276 | 2,874 | 182 |
| **# Test data:** | 9,767 | 7,535 | 2,480 | 180 |

Table 7: CelebA Dataset.

| **Target**: logical relationship; **Spurious feature**: presence of negation. | | | | | | |
| --- | --- | --- | --- | --- | --- | --- |
| **Group $g$:** | 0 | 1 | 2 | 3 | 4 | 5 |
| **Target $y \in \{0, 1, 2\}$:** | 0 (con) | 0 (con) | 1 (ent) | 1 (ent) | 2 (neu) | 2 (neu) |
| **Spurious $s$:** | 0 (no neg) | 1 (neg) | 0 (no neg) | 1 (neg) | 0 (no neg) | 1 (neg) |
| **# Train data:** | 57,498 | 11,158 | 67,376 | 1,521 | 66,630 | 1,992 |
| **# Val data:** | 22,814 | 4,634 | 26,949 | 613 | 26,655 | 797 |
| **# Test data:** | 34,597 | 6,655 | 40,496 | 886 | 39,930 | 1,148 |

Table 8: MultiNLI Dataset.

- JTT (Liu et al., 2021) first trains a standard ERM model and identifies misclassified training examples, then retrains a second model with these examples upweighted. A validation set with group labels is used to tune the number of epochs in the first stage and the upweighting factor in the second.

- AFR (Qiu et al., 2023) adopts the same last-layer retraining paradigm: after ERM training, it retrains the final layer on a held-out split using a loss-weighted objective that prioritizes poorly predicted examples. Group labels are not needed for retraining and are used only on the validation set for hyperparameter tuning and model selection.

- DaC (Noohdani et al., 2024) improves group robustness via last-layer retraining. It first performs ERM training, then applies adaptive masking and uses the masked images for data augmentation; the augmented data are used to retrain the final layer. As with JTT, group labels are used only on the validation set for hyperparameter tuning and model selection.

JTT, AFR, and DaC rely only on group-labeled validation data for tuning and model selection. They do not require group labels during training. These methods are representative group-label-efficient approaches with strong worst-group performance, making them ideal baselines for evaluating the effectiveness of SIEVE.

### B.1.3 HYPERPARAMETER AND HARDWARE CONFIGURATIONS

**Common Training Setup.** For all experiments on vision datasets, we train a ResNet-50 model pretrained on ImageNet using SGD with a momentum of 0.9. Results reported over 5 trials are averaged across 5 random seeds (1–5), and those over 3 trials are averaged across seeds 1–3. For MultiNLI, we use the HuggingFace implementation of the BERT model, initialized with pretrained weights (Devlin et al., 2019). We fine-tune the model for 10 epochs using the AdamW optimizer with a batch size of 32, no weight decay, and an initial learning rate of $2 \times 10^{-5}$.

**SIEVE Configuration.** For all methods that are combined with SIEVE, we set $N_c$ equal to $N_{nc}$. Specifically, for Dominoes, Waterbirds and CelebA, $N_c = 200$ and $T_{iter} = 20$; for Metashift, $N_c = 100$ and $T_{iter} = 10$; for MultiNLI, $N_c = 8,000$ and $T_{iter} = 5$.

In Algorithm 1, the parameter $K$ denotes the number of spurious and non-spurious examples selected from the confusing set. In our implementation, instead of specifying $K$ directly, we adopt a percentile-based selection strategy based on loss change dynamics. We set the quantile threshold $\tau = 0.8$ in all experiments, which corresponds to selecting the top 20% of samples with the largest loss increase (for non-spurious selection) and the top 20% of samples with the fastest loss decrease (for spurious selection). This approach provides a flexible approximation to $K$, adapting to dataset-specific training behavior.

**Method-Specific Settings.**

- **ERM and ERM-SIEVE.** The learning rate is set to 0.001 for all vision datasets. Weight decay is set to 0.001 for Dominoes, Waterbirds, and Metashift, and 0.0001 for CelebA. For Dominoes, we use a batch size of 16, train for 15 epochs, and do not apply any data augmentation. For Waterbirds, we use a batch size of 32 and train for 100 epochs. For Metashift, we use a batch size of 16 and train for 100 epochs. For CelebA, we use a batch size of 128 and train for 30 epochs.

- **JTT and JTT-SIEVE.** For JTT, we tune JTT-specific hyperparameters following the original JTT paper (Liu et al., 2021). Specifically, we use:
    - Waterbirds: epoch 60, upweighting factor 100
    - Dominoes: epoch 40, upweighting factor 50
    - Metashift: epoch 60, upweighting factor 50
    - CelebA: epoch 1, upweighting factor 50
    - MultiNLI: epoch 2, upweighting factor 6

  For Dominoes and Metashift, we use a learning rate of 0.001 and a weight decay of 0.001. For Waterbirds and CelebA, we follow the original JTT paper (Liu et al., 2021). Specifically, for Waterbirds, we use a learning rate of 0.00001 and a weight decay of 1; for CelebA, we use a learning rate of 0.00001 and a weight decay of 0.1.

  For JTT-SIEVE on Waterbirds, we use a learning rate of 0.0001, a weight decay of 0.001, 50 epochs, and an upweighting factor of 50. For CelebA, we use 5 epochs for computational efficiency, while keeping all other settings the same. For MultiNLI, we use epoch 2, upweighting factor 4. For the remaining datasets, JTT-SIEVE uses the same settings as JTT.

- **AFR and AFR-SIEVE.** We follow AFR (Qiu et al., 2023) for hyperparameter tuning and model selection. AFR adopts last-layer retraining with a frozen backbone (Stage 1: ERM training; Stage 2: last-layer retraining with a loss-weighted objective). The validation set is used in Stage 2 to tune $\gamma$ and $\lambda$ and for model selection. Here, $\gamma$ controls the degree of upweighting for poorly predicted examples, and $\lambda$ is an $\ell_2$ regularization coefficient on the last-layer parameters during retraining.

  For Waterbirds and CelebA, we adopt the Stage 1 and Stage 2 hyperparameters from the original AFR implementation, while for MultiNLI we follow the hyperparameter search protocol described in the AFR paper:
    - Waterbirds: Stage 1: batch size 32, learning rate 0.003, weight decay 0.0001, 50 epochs. Stage 2: $\gamma = 10$, $\lambda = 0.0$, 500 epochs.
    - CelebA: Stage 1: batch size 32, learning rate 0.003, weight decay 0.0001, 20 epochs. Stage 2: $\gamma = 0.001$, $\lambda = 1.44$, 1000 epochs.
    - MultiNLI: Stage 1: batch size 32, learning rate 0.00002, weight decay 0, 5 epochs. Stage 2: learning rate 0.01, 200 epochs; we sweep $\gamma$ over 10 values linearly spaced in $[10^2, 10^5]$ and $\lambda$ over 26 values linearly spaced in $[0, 50]$.

  For AFR-SIEVE on these three datasets, we tune $\gamma$ and $\lambda$ on the SIEVE-constructed validation set while keeping other settings unchanged. We select: Waterbirds ($\gamma = 4.5$, $\lambda = 0.0$) and CelebA ($\gamma = 1.0$, $\lambda = 0.0$).

  For Dominoes and Metashift, Stage 1 follows DaC (Noohdani et al., 2024): learning rate 0.001 and weight decay 0.001 for both; batch size 16 with 15 epochs for Dominoes; batch size 16 with

100 epochs for Metashift. In Stage 2, we set batch size 32 and train for 500 epochs. We grid-search $\gamma$ over 33 values linearly spaced in $[4, 20]$ and $\lambda \in \{0, 0.1, 0.2, 0.3, 0.4\}$. The selected values are:

- AFR: Dominoes ($\gamma = 4.0$, $\lambda = 0.1$); Metashift ($\gamma = 6.0$, $\lambda = 0.0$).
- AFR-SIEVE: Dominoes ($\gamma = 4.0$, $\lambda = 0.4$); Metashift ($\gamma = 6.5$, $\lambda = 0.0$).

- **DaC and DaC-SIEVE.** We set all hyperparameters following the original DaC paper (Noohdani et al., 2024).

More specifically, during the ERM stage of DaC, we use a learning rate of $0.001$ and a weight decay of $0.001$ for Dominoes, Waterbirds, and Metashift, and a weight decay of $0.0001$ for CelebA. The batch size and number of training epochs are as follows:

- Waterbirds: batch size 32, 100 epochs
- Dominoes: batch size 16, 15 epochs
- Metashift: batch size 16, 100 epochs
- CelebA: batch size 128, 30 epochs

During the last-layer retraining stage of DaC, we use a learning rate of $0.005$ for all datasets. A step learning rate scheduler with a step size of 5 and gamma of $0.5$ is used. The batch size is set to $64$ for all datasets. No regularization terms are applied during the last-layer retraining.

The causal masking flag is set to `True` for the following datasets and hyperparameter settings:

- Waterbirds: 20 epochs, $\alpha = 10$, $q = 0.6$
- Dominoes: 20 epochs, $\alpha = 6$, $q = 0.8$
- Metashift: 30 epochs, $\alpha = 6$, $q = 0.5$
- CelebA: 15 epochs, $\alpha = 5$, $q = 0.2$

The flag is set to `False` for all other datasets.

**Hardware Setup.** All experiments were conducted on a server equipped with an NVIDIA RTX A6000 GPU (49GB VRAM), an AMD Ryzen Threadripper PRO 3975WX 32-core CPU (64 threads), and 125GB of RAM.

## C  ADDITIONAL ANALYSIS OF SIEVE

### C.1  LOSS DYNAMICS ANALYSIS

In Figure 1(b) and Figure 2, we illustrated our key observation on the Waterbirds dataset: within the group of confusing examples, spurious samples tend to exhibit a faster loss decrease in the early epochs, while non-spurious samples either decrease more slowly or even increase in loss. To verify whether this behavior generalizes beyond Waterbirds, we perform the same loss dynamics analysis on Dominoes, Metashift and CelebA. The results are shown in Figures 5 to 10.

**Dominoes.** Figure 5(a) shows that confusing examples exhibit higher training loss than non-confusing ones. Figure 5(b) further reveals a clear spurious vs. non-spurious loss separation within the confusing group in the early epochs. As shown in both Figure 5(b) and Figure 6, the training loss of spurious examples tends to decrease rapidly, while that of non-spurious examples decreases more slowly or even increases.

**Metashift.** In Figure 7(b) and Figure 8, we observe the same trend. Spurious confusing examples consistently show faster loss reduction, while non-spurious confusing examples exhibit a noticeable increase in loss during the early training stage.

**CelebA.** In Figure 9(b) and Figure 10, the same pattern is again observed. Confusing examples exhibit higher losses than non-confusing ones. Within the confusing category, spurious examples show a clear loss decrease, while non-spurious examples demonstrate a sharp loss increase during the early training stage.

**MultiNLI.** We observe consistent behavior on the NLP benchmark as well, as shown in Figure 11, that distinct trends emerge within the confusing category: spurious examples (contradiction pairs with negation) show a loss decrease, whereas non-spurious examples (entailment or neutral pairs with negation) exhibit a loss increase during the early training stage. This consistency across both vision and language tasks indicates that the loss dynamics we identify generalize beyond vision to language.

These results demonstrate that our observation is robust and generalizes across datasets with different characteristics.

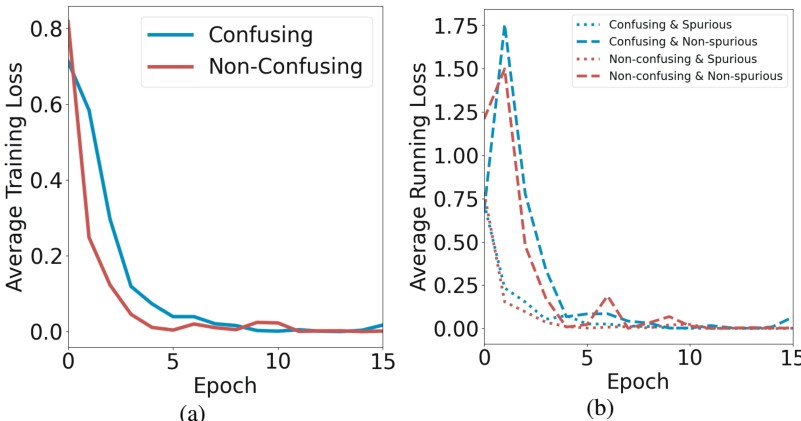

Figure 5: (a) Training losses of 100 confusing and 100 non-confusing examples from the Dominoes dataset. (b) The same examples in (a), further split by spurious and non-spurious groups.

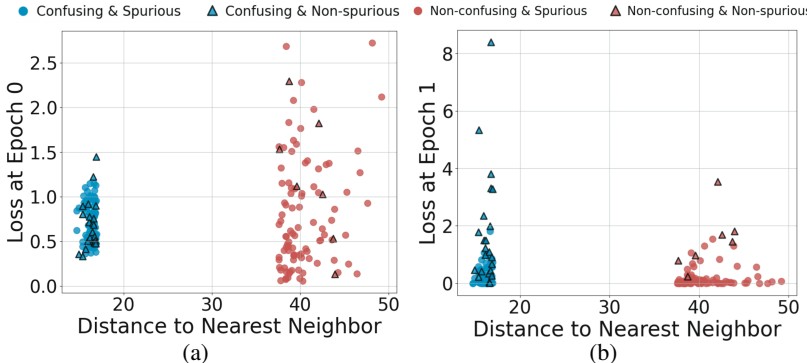

Figure 6: (a) The training loss of examples in four subgroups of Dominoes on epoch 0. (b) The training loss of examples in four subgroups of Dominoes on epoch 1. Here, the x-axis denotes the distance to the nearest other-class example. We take the 100 examples with the least distance as the confusing examples, and the 100 examples with the largest distance as the non-confusing ones, as done in Eq. (2) and Eq. (3) respectively, resulting in two sharp boundaries defined by two threshold distances values.

## C.2 COMPLETE HYPERPARAMETER SENSITIVITY RESULTS

As described in Section 3.3, we conducted comprehensive sensitivity analyses for SIEVE under different hyperparameter settings. Due to space limitations in the main paper, we present the full set of results here in Table 9. These results cover various combinations of $\tau \in \{0.8, 0.9\}$, $N_c \in \{100, 200, 300\}$, and $T_{\text{iter}} \in \{5, 10, 20\}$, evaluated on Waterbirds, Dominoes, and Metashift. All entries are averaged over three random seeds, except for the bolded results (reported in the main paper), which are averaged over five seeds.

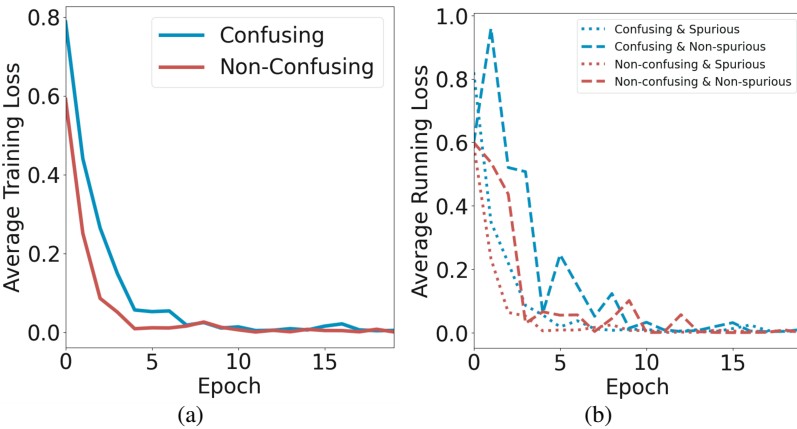

Figure 7: (a) Training losses of 100 confusing and 100 non-confusing examples from the Metashift dataset. (b) The same examples in (a), further split by spurious and non-spurious groups.

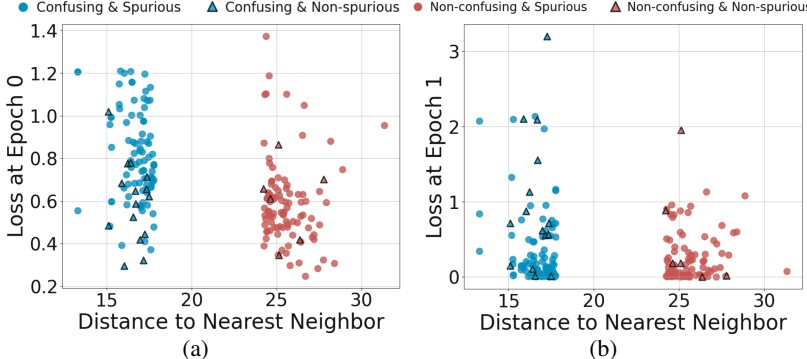

Figure 8: (a) The training loss of examples in four subgroups of Metashift on epoch $0$. (b) The training loss of examples in four subgroups of Metashift on epoch $1$. Here, the x-axis denotes the distance to the nearest other-class example. We take the 100 examples with the least distance as the confusing examples, and the 100 examples with the largest distance as the non-confusing ones, as done in Eq. (2) and Eq. (3) respectively, resulting in two sharp boundaries defined by two threshold distances values.

## C.3 WHY WE SELECT FROM CONFUSING EXAMPLES ONLY

To explore whether spurious and non-spurious examples can also be effectively separated within the non-confusing set, we visualize the distribution of loss change $\Delta L = L^{(1)} - L^{(0)}$ for both confusing and non-confusing examples in Figure 12. Across all four datasets, we compare how well these two subsets allow for distinguishing spurious from non-spurious examples: In each histogram, we examine the distributional differences between spurious and non-spurious examples; since our method determines spuriousness based on training loss change at the extremes—specifically, large loss increases (left) and large loss decreases (right)—we focus on whether these regions exhibit clear separation across different datasets.

On the Waterbirds dataset, the separation between spurious and non-spurious examples is relatively clear within the confusing group. Moreover, we observe that non-spurious examples are rare in the non-confusing set—most non-confusing examples are spurious. This suggests that while it is technically possible to extract some spurious examples from the non-confusing group, the confusing group remains the more informative and reliable source for both spurious and non-spurious samples.

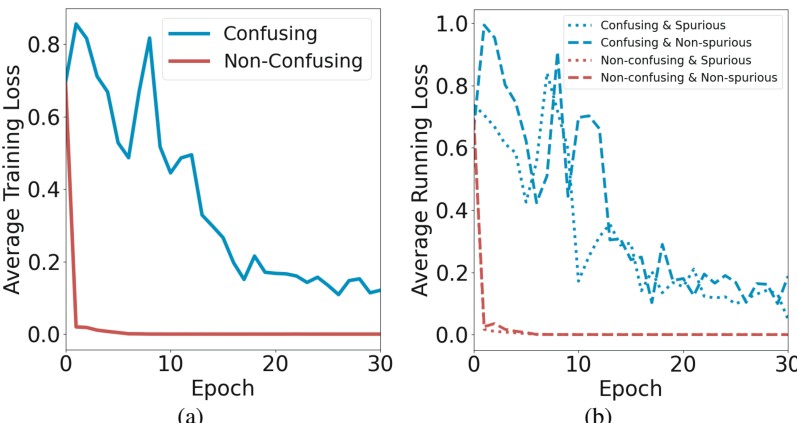

Figure 9: (a) Training losses of 100 confusing and 100 non-confusing examples from the CelebA dataset. (b) The same examples in (a), further split by spurious and non-spurious groups.

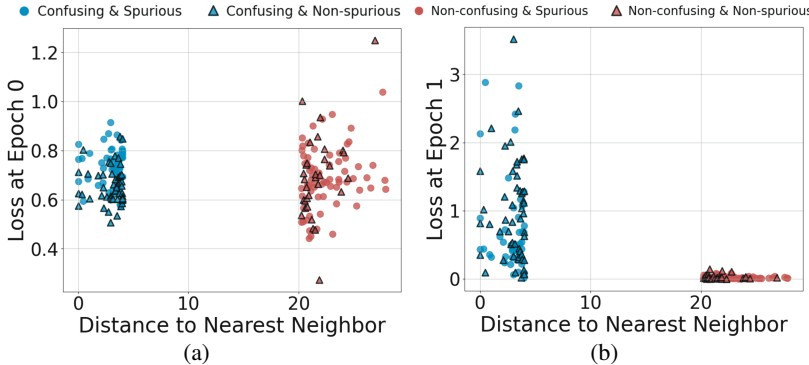

Figure 10: (a) The training loss of examples in four subgroups of CelebA on epoch 0. (b) The training loss of examples in four subgroups of CelebA on epoch 1. Here, the x-axis denotes the distance to the nearest other-class example. We take the 100 examples with the least distance as the confusing examples, and the 100 examples with the largest distance as the non-confusing ones, as done in Eq. (2) and Eq. (3) respectively, resulting in two sharp boundaries defined by two threshold distances values.

However, the same observation does not generalize well to the Dominoes and Metashift datasets. In Metashift, although the selection within the confusing group may include some mislabeled non-spurious examples, the separation remains reasonably effective. In contrast, distinguishing spurious and non-spurious examples within the non-confusing group proves challenging, with little reliable separation observed.

The difficulty is even more pronounced in Dominoes, where spurious and non-spurious examples in the non-confusing group are heavily mixed and indistinguishable based on loss dynamics, while the confusing group allows us to extract a cleaner set of spurious examples and a moderately accurate set of non-spurious examples.

The most challenging case is CelebA. From the non-confusing examples, it is nearly impossible to separate spurious from non-spurious examples in a reliable way. While the separation within the confusing group is less accurate than in Dominoes or Waterbirds, it still enables the extraction of much more accurate spurious examples and moderately reliable non-spurious examples—far better than using the non-confusing set.

These results support our design choice to select both spurious and non-spurious samples exclusively from the confusing group. While this strategy may occasionally introduce some noise in the pseudo

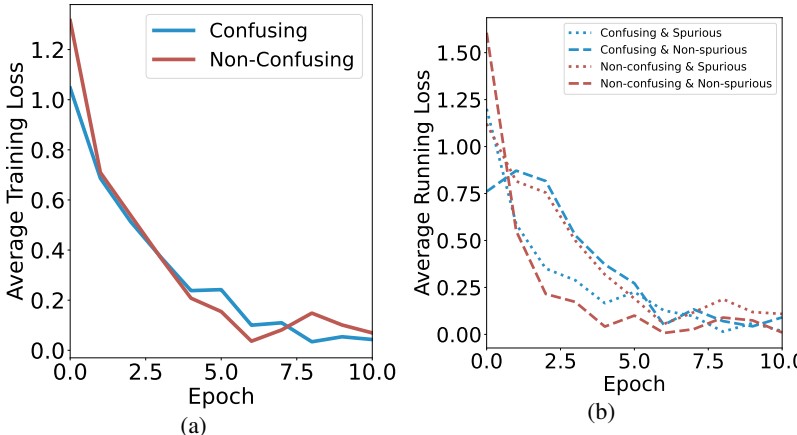

Figure 11: (a) Training losses of 100 confusing and 100 non-confusing examples from the MultiNLI dataset. (b) The same examples in (a), further split by spurious and non-spurious groups.

Table 9: We report the average worst-group accuracy (Worst) and average accuracy (Average) over 3 random seeds on each dataset under different SIEVE configuration. The **bold** numbers correspond to the results from the main experiments: $N_c = 200$, $T_{iter} = 20$, $\tau = 0.8$ for Waterbirds and Dominoes, and $N_c = 100$, $T_{iter} = 10$, $\tau = 0.8$ for Metashift. These values are copied from the 5-seed evaluations in Table 1 for consistency.

| Method | $N_c$ | $T_{iter}$ | $\tau$ | Waterbirds | | Metashift | | Dominoes | |
|---|---|---|---|---|---|---|---|---|---|
| | | | | Worst | Average | Worst | Average | Worst | Average |
| DaC | - | - | - | 92.3 (0.4) | 95.3 (0.4) | 78.30 (1.6) | 79.3 (0.1) | 89.2 (0.1) | 92.2 (0.3) |
| DaC-SIEVE ($\tau = 0.8$) | 100 | 5 | 0.8 | 89.72 (1.48) | 95.20 (0.17) | 78.32 (3.35) | 82.93 (2.24) | 87.01 (2.77) | 90.53 (0.86) |
| | 200 | 5 | 0.8 | 88.68 (1.96) | 94.66 (0.23) | 79.86 (2.46) | 83.61 (1.98) | 89.59 (1.14) | 92.24 (0.45) |
| | 300 | 5 | 0.8 | 89.72 (1.53) | 94.87 (0.36) | 78.65 (3.02) | 82.99 (2.00) | 86.12 (3.01) | 90.53 (0.16) |
| | 100 | 10 | 0.8 | 90.29 (1.71) | 94.80 (0.49) | **81.37 (0.60)** | 81.08 (2.38) | 89.11 (0.12) | 92.64 (0.16) |
| | 200 | 10 | 0.8 | 89.27 (1.02) | 93.95 (2.02) | 82.06 (0.68) | 83.98 (0.75) | 89.80 (0.89) | 92.35 (0.53) |
| | 300 | 10 | 0.8 | 88.84 (0.78) | 95.05 (0.20) | 81.63 (0.75) | 83.92 (0.64) | 89.87 (1.03) | 91.50 (0.74) |
| | 100 | 20 | 0.8 | 90.99 (0.97) | 93.83 (0.42) | 80.52 (1.44) | 82.07 (0.98) | 89.80 (0.89) | 92.35 (0.53) |
| | 200 | 20 | 0.8 | **91.71 (1.95)** | **94.06 (1.29)** | 81.48 (0.38) | 83.30 (1.98) | **89.59 (0.69)** | **92.35 (0.53)** |
| | 300 | 20 | 0.8 | 91.16 (1.13) | 93.86 (0.14) | 80.61 (2.47) | 82.44 (1.92) | 89.93 (0.72) | 92.47 (0.64) |
| DaC-SIEVE ($\tau = 0.9$) | 100 | 5 | 0.9 | 89.98 (2.03) | 94.82 (0.40) | 76.24 (6.53) | 83.73 (0.85) | 82.99 (4.60) | 88.93 (2.83) |
| | 200 | 5 | 0.9 | 89.72 (1.53) | 94.78 (0.45) | 74.12 (2.77) | 81.95 (2.68) | 84.63 (2.74) | 90.22 (0.61) |
| | 300 | 5 | 0.9 | 90.21 (1.67) | 94.50 (1.05) | 78.26 (2.25) | 81.82 (2.48) | 84.63 (2.74) | 90.22 (0.61) |
| | 100 | 10 | 0.9 | 90.50 (1.64) | 94.73 (0.51) | 81.81 (0.50) | 83.36 (0.67) | 89.32 (0.43) | 92.99 (0.61) |
| | 200 | 10 | 0.9 | 90.55 (1.56) | 94.76 (0.47) | 75.42 (5.82) | 80.71 (2.77) | 89.32 (0.43) | 92.99 (0.61) |
| | 300 | 10 | 0.9 | 90.24 (1.67) | 94.61 (0.68) | 79.66 (3.91) | 82.50 (2.14) | 90.11 (0.84) | 93.87 (1.77) |
| | 100 | 20 | 0.9 | 90.45 (0.99) | 94.43 (0.71) | 80.92 (1.94) | 84.75 (0.39) | 89.46 (1.33) | 92.35 (0.53) |
| | 200 | 20 | 0.9 | 90.71 (0.78) | 94.41 (0.52) | 81.66 (0.92) | 83.24 (2.21) | 89.80 (0.89) | 92.35 (0.53) |
| | 300 | 20 | 0.9 | 90.55 (1.10) | 94.74 (0.18) | 81.63 (0.75) | 83.92 (0.64) | 89.80 (0.89) | 92.35 (0.53) |

group labels, particularly in more challenging datasets like CelebA, it consistently offers higher labeling accuracy and stronger group separation than relying on the non-confusing set.

## C.4 RUNNING TIME

To better understand the computational overhead introduced by SIEVE, we report the running time (in minutes) for both the SIEVE selection phase and standard ERM training across four datasets in Table 10. While SIEVE incurs some additional computation, its cost remains modest compared to the full training time. Notably, on datasets like Metashift and Dominoes, the selection time is under 5 minutes, demonstrating the efficiency and practicality of our method.

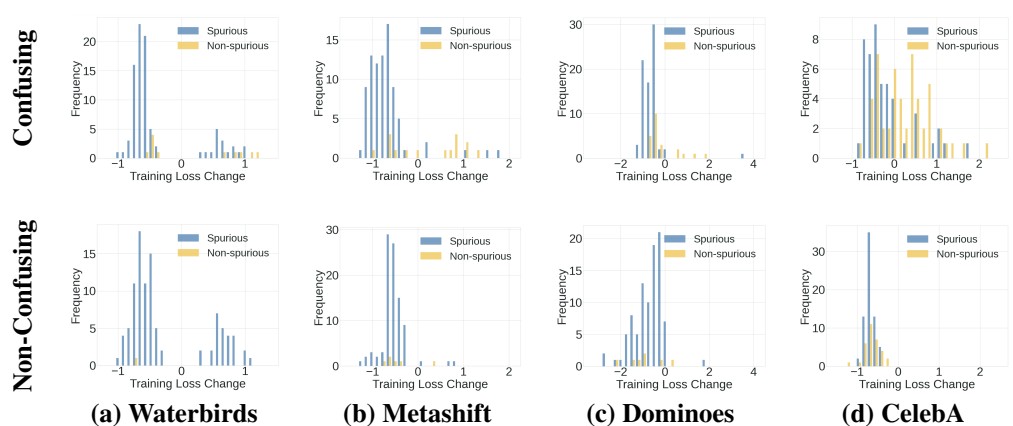

(a) Waterbirds      (b) Metashift      (c) Dominoes      (d) CelebA

Figure 12: Histograms of training loss change ($\Delta L = L^{(1)} - L^{(0)}$) for confusing and non-confusing examples across four datasets. The top row shows the confusing examples, while the bottom row shows the corresponding non-confusing examples.

Table 10: Average running time (in minutes) over 3 random seeds for SIEVE selection and ERM training on each dataset. Results are reported as mean (standard deviation).

| Dataset | SIEVE Selection Time | ERM Training Time |
|---|---|---|
| Waterbirds | 11.28 (0.04) | 82.17 (1.18) |
| Metashift | 2.22 (0.13) | 30.43 (1.30) |
| Dominoes | 4.36 (0.19) | 11.91 (0.20) |
| CelebA | 104.13 (0.22) | 426.96 (0.13) |

## C.5 LIMITATIONS

While SIEVE is simple and effective, it does have several limitations.

First, the accuracy of pseudo group labels can be low for underrepresented or ambiguous subgroups, particularly those with very limited data. Although the selected examples still tend to exhibit useful training dynamics, the group label noise may impact methods that rely heavily on precise group labels, such as JTT.

Second, SIEVE relies on the assumption that loss dynamics reflect spurious vs. non-spurious behavior. While this holds across the benchmarks that are commonly used in the literature, it may not generalize to other cases, such as data with high noise or extreme class imbalance.

Future work may focus on improving the robustness of pseudo label assignment in low-resource subgroups and exploring whether other indicators beyond training loss can offer more reliable separation between spurious and non-spurious examples.

## C.6 WHY JTT-SIEVE UNDERPERFORMS ON WATERBIRDS

In this section, we provide a more detailed discussion of why JTT-SIEVE underperforms on Waterbirds. At a high level, the reason is JTT is more sensitive to the errors in the constructed validation set, as in JTT, all the model parameters are tuned based on the validation set; in contrast, AFR and DaC are less sensitive because only the last-layer parameters are tuned.

JTT starts from a standard ERM model and then retrains the entire model in a second stage. It uses the validation set to choose (i) the first-stage training length $T$, (ii) the second-stage upweighting factor $\lambda$, and (iii) the early-stopping epoch in the second stage. By contrast, both AFR and DaC are last-layer retraining methods and there is no second stage in which a new full model is trained. Therefore, AFR-SIEVE and DaC-SIEVE achieve worst-group accuracies that are very close to those of AFR and DaC when using the same SIEVE pseudo-group information (see Table 1).

More specifically, AFR starts from a standard ERM model and retrains only the last layer with a weighted loss that emphasizes examples where the base model performs poorly, which implicitly upweights the minority group. The validation set is used to tune the reweighting strength and for early stopping. DaC is also a last-layer retraining method. In DaC, the validation set guides counterfactual data augmentation and early stopping: it uses worst-group accuracy to select (i) the fraction of samples per batch used to construct mixed images, (ii) the loss weight on mixed images, and (iii) whether to apply or invert the saliency mask. In both methods, the validation signal only affects a small number of hyperparameters on top of a fixed backbone, rather than driving the training of a new model from scratch. Consequently, AFR-SIEVE and DaC-SIEVE are much less sensitive than JTT-SIEVE when the available non-spurious signal is limited and noisy.

# D  SIMULATION STUDIES

## D.1  THE LOSS-DYNAMICS ASSUMPTION

In the following part, we instantiate the loss dynamics introduced in Section 2.1 in three controlled synthetic settings:

- First, we design a two-dimensional dataset with an explicitly visualizable causal and spurious feature under a high-prevalence, low-noise spurious setting in which the loss-dynamics assumption is expected to hold, and show that the loss dynamics on this dataset closely match the conceptual illustration in Figure 3.

- Second, we consider a three-dimensional construction with one causal feature and a single spurious feature, and systematically vary the prevalence and the noise of the spurious signal to understand when the loss-dynamics assumption holds or fails.

- Finally, we extend this construction to a four-dimensional setting with two independent spurious features, and study whether the same loss-dynamics separation persists in the presence of multiple spurious correlations.

### D.1.1  SYNTHETIC DATASET CONSTRUCTION.

We consider three synthetic datasets. In all of them, we train a logistic regression model using the same optimization procedure and the same definitions of confusing/non-confusing and spurious/non-spurious examples; only the construction (and hence dimensionality) of the causal and spurious features differs across datasets.

**2D dataset construction.**  Our first synthetic dataset is a two-dimensional toy example. For each example $i$, we draw a latent scalar $z_i \sim \mathcal{N}(0,1)$ and set the label $y_i = \text{sign}(z_i) \in \{-1,+1\}$. The causal feature is a squashed version of $z_i$, $x_i^{(\text{causal})} = \tanh(\alpha z_i)$ with $\alpha = 0.7$.

We then introduce a binary spurious attribute $a_i \in \{-1,+1\}$ that agrees with the label with probability $p_{\text{sp}}$, i.e., $a_i = y_i$ with probability $p_{\text{sp}}$ and $a_i = -y_i$ otherwise. We refer to this probability as the *prevalence* of the spurious attribute in the training data.

The observed spurious feature is a noisy version of this attribute,

$$x_i^{(\text{spu})} = a_i \mu_{\text{sp}} + \epsilon_i, \qquad \epsilon_i \sim \mathcal{N}(0, \sigma_{\text{sp}}^2), \tag{7}$$

with $\mu_{\text{sp}} = 1$ and $\sigma_{\text{sp}}$ controlling the noise level. The final input is two-dimensional, $\mathbf{x}_i = (x_i^{(\text{spu})}, x_i^{(\text{causal})}) \in \mathbb{R}^2$. We refer to examples with $a_i = y_i$ as *spurious*, and to those with $a_i = -y_i$ as *non-spurious*. In our visualizations (Figure 13), we instantiate this construction with $p_{\text{sp}} = 0.9$ and $\sigma_{\text{sp}} = 0$, where the spurious feature is dominant and noiseless.

**3D dataset with a single spurious feature.**  To study when the loss dynamics holds and fails, we construct our second sythetic dataset $\{(\mathbf{x}_1, y_1), \ldots, (\mathbf{x}_n, y_n)\}$, where each input-output pair $(\mathbf{x}_i, y_i) \in \mathbb{R}^3 \times \{-1,+1\}$ is sampled from the underlying data distribution as follows. First, the label $y_i$ is determined by an unobserved latent feature $z_i$:

$$y_i = \text{sign}\big(\sin(5z_i) + \exp(z_i)\big), \qquad z_i \sim \text{Unif}(-1,1). \tag{8}$$

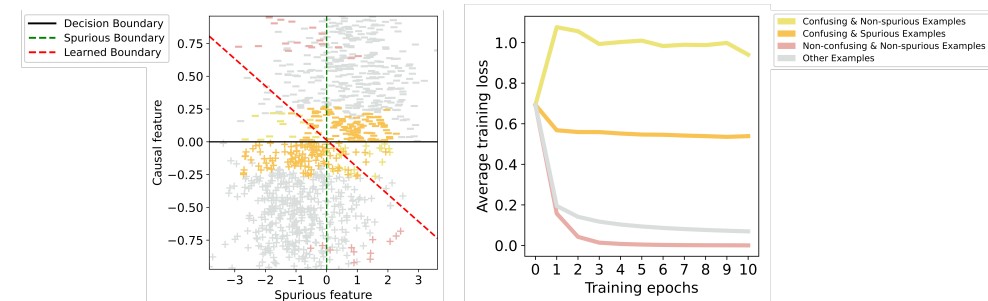

Figure 13: Two-dimensional illustrative synthetic dataset. **Left:** A subset of $n = 1{,}000$ training examples with the true decision boundary (horizontal line), the spurious boundary (vertical line), and the learned linear classifier. **Right:** Training loss over epochs on the full synthetic dataset ($n = 10{,}000$) for the four groups. Confusing spurious examples show a fast early loss drop, while confusing non-spurious examples improve much more slowly, matching the loss-dynamics pattern in Figure 3.

The observed features $\mathbf{x}_i = (x_i^{(1)}, x_i^{(2)}, x_i^{(3)})$ consist of a causal feature $x_i^{(1)}$ and two spurious features $x_i^{(2)}$ and $x_i^{(3)}$, defined by

$$x_i^{(1)} = \tanh\big(\sin(5z_i) + \exp(z_i)\big), \quad x_i^{(2)} = b_i \sin(5z_i) + \epsilon_i^{(2)}, \quad x_i^{(3)} = b_i \exp(z_i) + \epsilon_i^{(3)}, \quad (9)$$

where $b_i$ takes value $+1$ with probability $p_{\mathrm{sp}}$ and $-1$ otherwise, and $\epsilon_i^{(2)}$ and $\epsilon_i^{(3)}$ are sampled independently from $\mathcal{N}(0, \sigma_{\mathrm{sp}}^2)$. Example $i$ is *spurious* if $\mathrm{sign}\big(x_i^{(2)} + x_i^{(3)}\big) = y_i$ and *non-spurious* otherwise.

**4D dataset with two spurious features.** To study the effect of multiple spurious correlations, we extend the above three-dimensional construction by adding a second, independent spurious feature. We keep the same latent variable $z_i \sim \mathrm{Unif}(-1, 1)$, label $y_i = \mathrm{sign}(\sin(5z_i) + \exp(z_i))$, and features $x_i^{(1)}, x_i^{(2)}, x_i^{(3)}$ defined exactly as above, with the same parameters $p_{\mathrm{sp}}$ and $\sigma_{\mathrm{sp}}$ governing the first spurious feature $(x_i^{(2)}, x_i^{(3)})$. We then define an additional spurious feature $x_i^{(4)}$ by

$$x_i^{(4)} = b_i^{(4)}\big(-2\big[\sin(5z_i) + \exp(z_i)\big]\big) + \epsilon_i^{(4)}, \quad (10)$$

where $b_i^{(4)} \in \{+1, -1\}$ takes value $+1$ with probability $p_{\mathrm{sp}}^{(4)}$ and $-1$ otherwise, and $\epsilon_i^{(4)} \sim \mathcal{N}(0, (\sigma^{(4)})^2)$. The full input vector is $\mathbf{x}_i = (x_i^{(1)}, x_i^{(2)}, x_i^{(3)}, x_i^{(4)}) \in \mathbb{R}^4$.

In this four-dimensional construction, we declare an example "spurious" if either the first spurious feature $\mathrm{sign}(x_i^{(2)} + x_i^{(3)})$ or the second spurious feature $\mathrm{sign}(-x_i^{(4)})$ predicts the label correctly. This models a setting with multiple, partially overlapping spurious correlations that the classifier can exploit.

### D.1.2 EXPERIMENTAL SETUP.

Across all three synthetic constructions, we use the same model, training procedure, and way of defining confusing/non-confusing and spurious/non-spurious examples.

**Model and optimization.** In all experiments, we generate $10{,}000$ examples and train a logistic regression model on these examples. The model takes the full feature vector $\mathbf{x}_i$ as input and is optimized with stochastic gradient descent (learning rate $5 \times 10^{-1}$, batch size 128) for 10 epochs. The weights are initialized to zero, so that the initial predictor is completely uninformative and any early loss dynamics are driven purely by the data rather than by random initialization.

**Confusing and non-confusing examples.** For each synthetic dataset, we define confusing and non-confusing examples based on $d_i$ defined in Eq. (1) the same as in our real-data experiments. We sort the training examples by $d_i$ and take the bottom $20\%$ (smallest $d_i$) as *confusing* and the top $20\%$ (largest $d_i$) as *non-confusing*.

**Loss dynamics and summary statistic.** For each training epoch $t$, we track the average training loss in each of the four groups. As introduced in Section 2.2, we focus on the early loss change $\Delta L_i^{(t_0)}$ after one epoch of training ($t_0 = 1$ in Algorithm 1). Let $\mathcal{C}$ denote the set of confusing examples. We summarize the separation between confusing non-spurious and confusing spurious examples by

$$\Delta_{\mathrm{conf}} := \mathbb{E}\big[\Delta L_i^{(t_0)} \,\big|\, i \in \mathcal{C}, \ i \text{ non-spurious}\big] - \mathbb{E}\big[\Delta L_i^{(t_0)} \,\big|\, i \in \mathcal{C}, \ i \text{ spurious}\big], \tag{11}$$

where larger values of $\Delta_{\mathrm{conf}}$ indicate a stronger gap between confusing non-spurious and confusing spurious examples. In the figures below, we visualize the group-wise loss curves and report $\Delta_{\mathrm{conf}}$ as an additional scalar summary of the loss separation.

### D.1.3 RESULTS ON THE 2D DATASET.

With the two-dimensional construction, we verify that the loss-dynamics pattern in Figure 3 can indeed arise on a concrete dataset. We fix $p_{\mathrm{sp}} = 0.9$ and $\sigma_{\mathrm{sp}} = 0$, so that the spurious feature is highly reliable, and generate $n = 10{,}000$ training examples.

Figure 13 summarizes the results. On the left, we plot a subset of $n = 1{,}000$ examples together with the true decision boundary (horizontal line), the spurious boundary (vertical line), and the learned linear classifier. Confusing examples lie close to the true boundary, while non-confusing examples are farther away. Among them, spurious examples are located near the spurious boundary, whereas non-spurious examples are farther from it. On the right, the training loss curves show that confusing spurious examples exhibit a loss drop after the first epoch, whereas confusing non-spurious examples improve much more slowly and even experience a loss increase before eventually decreasing. Non-confusing examples maintain low loss throughout. This behavior matches the loss-dynamics pattern in Figure 3 and is consistent with what we observe on the five benchmarks in Figure 1 and section C.1.

### D.1.4 RESULTS ON THE 3D DATASET.

**Varying the prevalence and the noise level of the single spurious feature.** We first study when the loss-dynamics holds or fails on the constructed three dimensional synthetic dataset. To systematically vary the strength of the spurious feature, that is, how strongly the model can rely on it during training, we control two quantities: (i) the prevalence of the spurious examples $p_{\mathrm{sp}}$ and (ii) the noise level $\sigma_{\mathrm{sp}}$ of this spurious feature. The prevalence $p_{\mathrm{sp}}$ determines how often the spurious feature co-occurs with the label in the training data, whereas $\sigma_{\mathrm{sp}}$ controls how informative this feature remains when it is present. Together, these two factors determine how easy or difficult it is for the model to exploit the spurious correlation, which is the focus of our analysis. We consider a grid of configurations with $p_{\mathrm{sp}} \in \{0.9, 0.8, 0.7, 0.5\}$ and $\sigma_{\mathrm{sp}} \in \{0, 0.5, 1.0, 2.0\}$, ranging from clean and highly predictive spurious features to heavily corrupted ones. For each $(p_{\mathrm{sp}}, \sigma_{\mathrm{sp}})$, we generate $n = 10{,}000$ examples, train logistic regression as described above, and plot the average loss over epochs for the four groups (confusing & spurious, confusing & non-spurious, non-confusing & spurious, non-confusing & non-spurious). The resulting $4 \times 4$ grid of loss curves is shown in Figure 14.

When $p_{\mathrm{sp}}$ is high and $\sigma_{\mathrm{sp}}$ is small (top-left region of Figure 14), confusing & spurious examples exhibit a rapid loss drop, whereas confusing & non-spurious examples improve much more slowly or even show a slight increase in loss, closely matching the conceptual illustration in Figure 3 and what we observe on the 2-D synthetic dataset in Figure 13. As $p_{\mathrm{sp}}$ decreases or $\sigma_{\mathrm{sp}}$ increases, the gap between the two curves within the confusing region becomes visibly smaller, and in the bottom-right panels (low $p_{\mathrm{sp}}$ and large $\sigma_{\mathrm{sp}}$) the spurious and non-spurious confusing examples follow very similar loss trajectories with no clear separation.

To summarize this trend, Figure 15 plots $\Delta_{\mathrm{conf}}(t_0 = 1)$ from equation 11 as a function of $p_{\mathrm{sp}}$ for several fixed values of $\sigma_{\mathrm{sp}}$. For small noise, $\Delta_{\mathrm{conf}}$ is large and positive when $p_{\mathrm{sp}}$ is high, and decreases as $p_{\mathrm{sp}}$ approaches 0.5. For larger noise levels, $\Delta_{\mathrm{conf}}$ is smaller across all $p_{\mathrm{sp}}$ and quickly shrinks towards zero. Taken together, these plots make explicit that the loss-dynamics separation is strongest when the spurious pattern is both prevalent and clean, and gradually disappears as either its prevalence or its reliability is reduced.

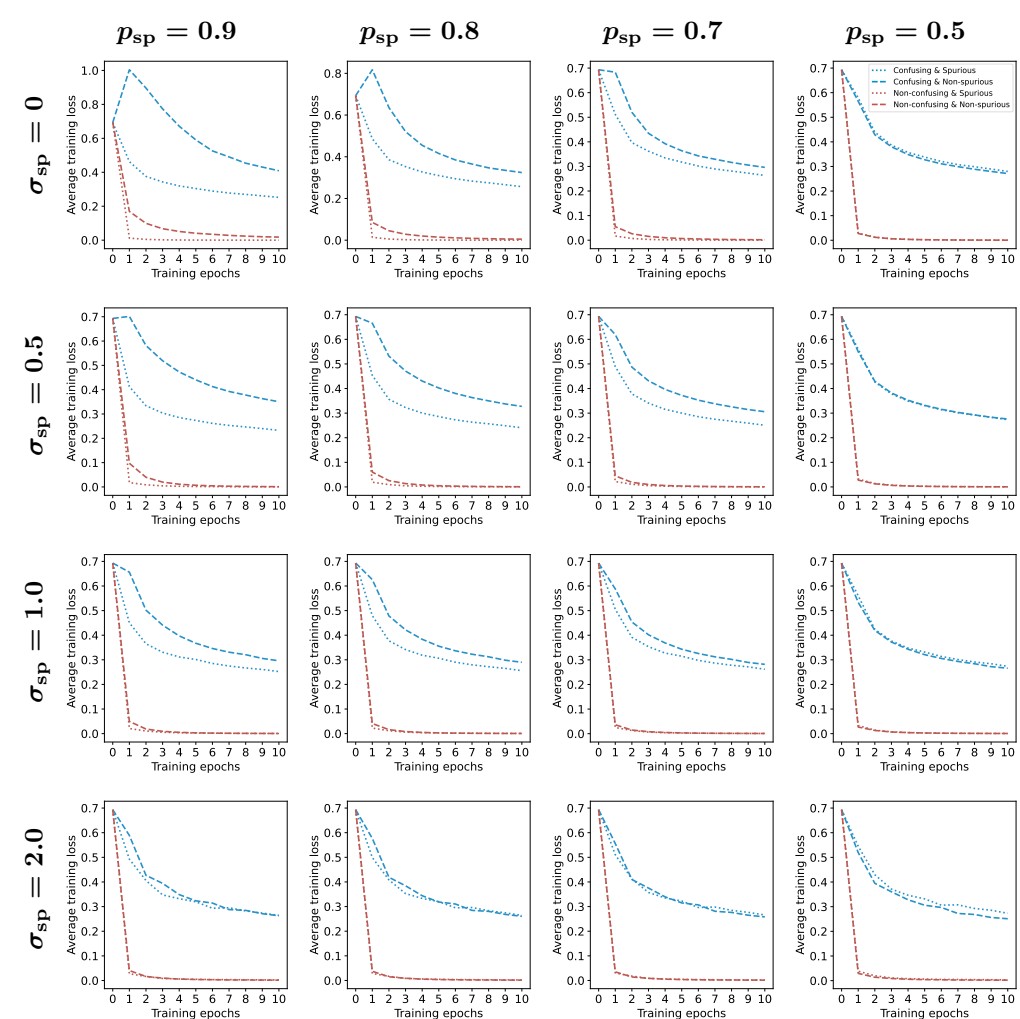

Figure 14: Average training loss over epochs on the synthetic dataset for different combinations of spurious prevalence $p_{\mathrm{sp}}$ (columns) and spurious noise level $\sigma_{\mathrm{sp}}$ (rows). Each panel shows the four groups from Figure 3: Confusing & Spurious, Confusing & Non-spurious, Non-confusing & Spurious, and Non-confusing & Non-spurious. The loss-dynamics separation within confusing examples is strongest when the spurious pattern is both prevalent and clean (top-left region), and gradually vanishes as $p_{\mathrm{sp}}$ decreases or $\sigma_{\mathrm{sp}}$ increases.

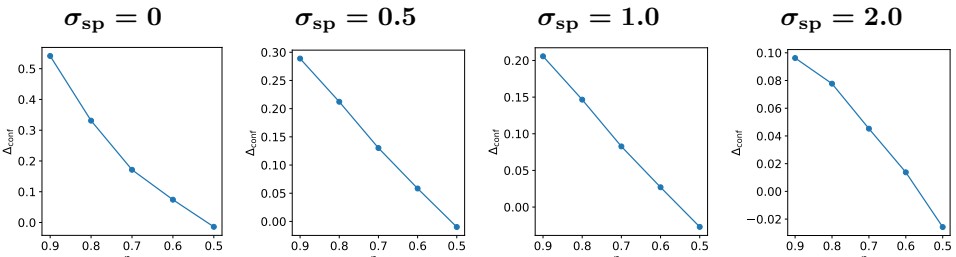

Figure 15: Separation in early loss dynamics on the synthetic dataset. For each spurious noise level $\sigma_{\mathrm{sp}}$, we plot the gap $\Delta_{\mathrm{conf}}(t_0 = 1)$ in equation 11. When the spurious pattern is both prevalent and clean (small $\sigma_{\mathrm{sp}}$), $\Delta_{\mathrm{conf}}$ is large and positive, indicating a strong separation in loss dynamics. As $\sigma_{\mathrm{sp}}$ increases, the effective strength of the spurious features decreases and $\Delta_{\mathrm{conf}}$ quickly shrinks towards zero.

### D.1.5 RESULTS ON THE 4D DATASET.

**Two spurious features in the four-dimensional dataset.** Finally, we consider the four-dimensional construction with two independent spurious features: one based on the pair $(x_i^{(2)}, x_i^{(3)})$ and one based on the additional feature $x_i^{(4)}$. In this setting, an example is labeled as spurious if either $\text{sign}(x_i^{(2)} + x_i^{(3)})$ or $\text{sign}(-x_i^{(4)})$ predicts its label correctly, as described in the construction above. We fix the first spurious features $(x_i^{(2)}, x_i^{(3)})$ to with $p_{\text{sp}} = 0.8$ and $\sigma_{\text{sp}} = 0$, and the prevalence of the second spurious feature to $p_{\text{sp}}^{(4)} = 0.8$. We then vary the noise level $\sigma^{(4)} \in \{0, 0.5, 1.0, 2.0\}$, while keeping all other settings unchanged, and plot the loss dynamics for the four groups in each case (Figure 16).

Across all noise levels, the separation between confusing spurious and confusing non-spurious examples remains largely stable: confusing spurious examples consistently demonstrate a larger early loss drop than confusing non-spurious ones, with only mild deviations when $\sigma^{(4)}$ is very large (the confusing & spurious curve becomes less smooth due to the increased randomness within this group). In contrast, the main effect of increasing $\sigma^{(4)}$ appears in the non-confusing examples. When $\sigma^{(4)} = 0$, the loss on non-confusing spurious examples decreases noticeably faster than on non-confusing non-spurious examples, since the model can additionally rely on the extra clean spurious feature. As $\sigma^{(4)}$ increases, the non-confusing spurious curve gradually moves closer to the non-confusing non-spurious curve, indicating that the noisy second spurious feature becomes harder to exploit and its influence on easy examples becomes much weaker.

Together with the experiments varying the prevalence above, these results suggest that our loss-dynamics assumption is robust to the presence of multiple spurious directions and SIEVE can be applied without modification in such settings.

## E ALTERNATIVE DEFINITIONS OF CONFUSINGNESS

We ran an additional experiment on Waterbirds where we defined confusing examples using prediction entropy under a trained ERM model instead of the pretrained-feature distance. We use an ERM model trained with the same architecture and hyperparameters as in our main Waterbirds experiments (a ResNet-50 pretrained on ImageNet and fine-tuned on Waterbirds; see Section B.1 for training details). We ran this trained ERM model on each example, used the prediction entropy (computed from its softmax probabilities) as a confusion score, and treated the highest-entropy examples as confusing. In this setting, the loss dynamics observed on our benchmarks and synthetic datasets no longer appear. In contrast, both confusing & spurious and confusing & non-spurious examples show a rapid loss increase at the beginning, and the non-spurious examples achieve higher loss later, while the spurious examples show a quicker and sharper loss increase early in training (see Figure 17), so the loss separation that SIEVE exploits largely disappears and SIEVE is not expected to work well.

Our confusingness score was designed to depend only on general, task-agnostic features, which are independent of any model trained on the current training set that contains spurious correlations. This is because if we rely on models trained on spurious data, then examples that are inherently confusing may no longer appear confusing in the model's learned representation space. The experiment on the alternative confusingness score below confirms that this can happen.

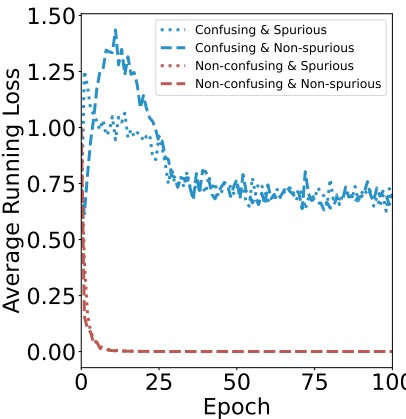

Figure 17: Loss dynamics on Waterbirds when confusingness is defined by prediction entropy. The two curves become almost indistinguishable, indicating that the entropy-based definition fails to produce the clear loss separation that SIEVE relies on.

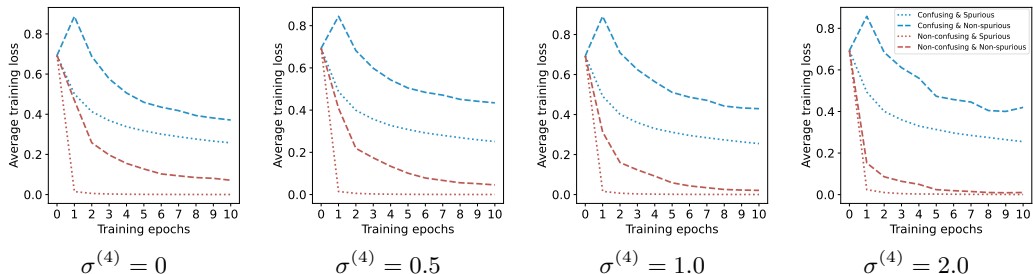

Figure 16: Four-dimensional construction with two spurious features. We fix the first spurious feature $(x^{(2)}, x^{(3)})$ (strong and noiseless) and $p_{\text{sp}}^{(4)} = 0.8$, and vary $\sigma^{(4)} \in \{0, 0.5, 1.0, 2.0\}$ for $x^{(4)}$. Confusing spurious examples consistently demonstrate a larger early loss drop, while non-confusing spurious examples gradually behave like non-confusing non-spurious ones as $\sigma^{(4)}$ increases.

## F STABILITY OF SIEVE WITH RESPECT TO DATA ORDERING

SIEVE uses the one-epoch loss change $\Delta L$ only to rank training examples; thus, its practical stability is reflected directly in the stability of the selected validation set. To assess the stability of SIEVE with respect to data ordering, we reuse the three random seeds from our Waterbirds experiments.

For each seed, we compare the resulting sets pairwise. Table 11 reports the Jaccard similarity and percentage overlap between the selected validation sets for each pair of seeds. We observe Jaccard similarities above 0.8 and overlaps around 90% across all seed pairs, indicating that $\Delta L$ produces a consistent ranking and is not driven by mini-batch ordering randomness.

Table 11: Stability of SIEVE validation sets on Waterbirds across three random seeds.

| Seed pair | (1, 2) | (1, 3) | (2, 3) |
|---|---|---|---|
| Jaccard similarity | 0.83 | 0.80 | 0.84 |
| Overlap (%) | 90.49 | 89.00 | 91.09 |

## G THE USE OF LLMS

We used LLMs solely for polishing the writing of the paper, including grammar correction and phrasing alternatives for clarity and brevity. All LLM-suggested edits were reviewed and verified by the authors, and all technical content is author-generated and author-validated.

