# OpenReview forum: "Group-Label-Free Validation for Spuriously Correlated Data"
_ICLR.cc/2026/Conference — Submitted to ICLR 2026_

### Official Review · Reviewer_YafZ · 2025-10-29

**Soundness:** 2
**Presentation:** 2
**Contribution:** 2
**Rating:** 2
**Confidence:** 4

**Summary:**

The paper proposes SIEVE (Spurious-aware IterativE Validation Example selection), a framework for building group-label-free validation sets to evaluate and tune models under spurious correlations. SIEVE identifies “confusing” examples close to the decision boundary in feature space, and then distinguishes between spurious and non-spurious examples based on their loss dynamics during early training: spurious examples tend to have rapidly decreasing losses, while non-spurious ones decrease more slowly or increase. By iteratively labeling these examples and aggregating them into a pseudo group-labeled validation set, SIEVE enables model selection without true group annotations. The method integrates easily into existing training pipelines such as ERM, JTT, AFR, and DaC. Experiments on Waterbirds, Dominoes, Metashift, and CelebA show that models validated with SIEVE perform comparably to those using real group-labeled validation sets.

**Strengths:**

1. The paper tackles a well-recognized and practical gap: robust model selection under spurious correlations when group labels are unavailable.

2. Compared to some heavy reweighting or adversarial methods, SIEVE’s additional cost (one short training epoch and distance computations) is modest.

**Weaknesses:**

1. There are already several related papers addressing the same goal of mitigating spurious correlations without group supervision [1, 2]. Moreover, as shown in Table 1, ERM-SIEVE underperforms compared to several baseline methods without SIEVE, raising questions about its practical effectiveness.

2. SIEVE selects confusing samples based on nearest neighbor distances in a pretrained feature space. However, the interpretation of “confusing” versus “non confusing” examples is heuristic and under justified.

3. The reliance on a single epoch to measure loss change seems arbitrary. Since the authors only train one epoch when computing the per sample loss change ΔL, the results may depend on the order of samples or mini batches rather than any intrinsic signal of spuriousness. The paper does not analyze or control for this randomness, so the method’s stability under different data orderings is uncertain.

4. The empirical gains reported in Table 1 are small and inconsistent. In many dataset and method combinations, SIEVE variants nothing or even underperform their baselines, often within one standard deviation. If I understand correctly, the paper does not include any baseline results where neither training nor validation uses group labels in Table 1. The non-SIEVE baselines still depend on group-labeled validation sets for hyperparameter tuning.

[1] Ghaznavi et al., “Trained Models Tell Us How to Make Them Robust to Spurious Correlation without Group Annotation” https://arxiv.org/abs/2410.05345

[2] Kirichenko et al., “Last Layer Re-Training is Sufficient for Robustness to Spurious Correlations” https://arxiv.org/abs/2204.02937

**Questions:**

1. Regarding Figure 2: The boundary between confusing and non-confusing examples appears unrealistically sharp. Could you explain in detail how Figure 2 was generated, including what metric or threshold was used to separate these two groups and whether any preprocessing or normalization influenced this clarity?

2. Regarding Figure 3: Shouldn’t there also be non-confusing but spurious examples in the figure? It seems that these cases are missing or mislabeled. Could you clarify on this?

3. Regarding Lines 185-187: The statement “For the non-confusing examples, they are relatively easy to classify, thus whether they are spurious or non-spurious, both the true decision boundary and the spurious decision boundary may classify them correctly” seems incorrect. If the classification relies on the spurious decision boundary, then non-confusing non-spurious examples would still be misclassified. Could you clarify?

---

> ### Author Response · Authors · 2025-11-30
> **Response to Reviewer YafZ (Part 1/2)**
>
> We thank the reviewer for acknowledging that SIEVE incurs only modest additional cost compared to heavier methods.
>
> ---
> ### Response to W1
>
>
> **Positioning vs. prior work [1,2]**
>
> We thank the reviewer for pointing out the related work [1]. As discussed in our paper, DFR [2] is not fully group-label-free because it requires a small group-balanced set for last-layer retraining. EVaLS [1] is fully group-label free last-layer-retraining strategy.
>
> Our work plays a different and complementary role: SIEVE is a plug-and-play mechanism that keeps existing training algorithms (ERM, JTT, AFR, DaC) unchanged and simply replaces their group-labeled validation sets with group-label-free ones. This makes SIEVE compatible with, and able to benefit from the strengths of many existing methods; for instance, combining DaC with SIEVE yields higher worst-group accuracy than EVaLS on Waterbirds (92.3 vs. 88.4).
>
> **In Table 1, ERM-SIEVE underperforms several baselines without SIEVE**
>
> We clarify that SIEVE is a plug-in module whose purpose is to remove the need for group labels in existing pipelines. All the baselines without SIEVE in Table 1 rely on group-labeled validation set, including ERM. So their higher performance is expected and not the comparison SIEVE is designed for.
>
> ---
> [1] Ghaznavi et al., “Trained Models Tell Us How to Make Them Robust to Spurious Correlation without Group Annotation” https://arxiv.org/abs/2410.05345
>
> [2] Kirichenko et al., “Last Layer Re-Training is Sufficient for Robustness to Spurious Correlations” https://arxiv.org/abs/2204.02937
>
> ---
>
> ### Response to W2
>
>
>
> **The interpretation of "confusing" versus "non confusing" examples is heuristic and under justified**
>
> We would like to clarify that "confusing" and "non-confusing" are not introduced as new theoretical constructs, but as descriptive terms for a natural geometric phenomenon in representation space: a sample is "confusing" when it lies close to examples from other classes, and "non-confusing" when its local neighborhood is aligned with its class, as defined in Eqs. (2)–(3)  in the paper. The terminology is purely operational and could be replaced with any equivalent wording; it does not rely on any additional assumptions beyond the observed feature and label space geometry.
>
>
>
> ---
>
>
> ### Response to W3:
>
>
>
> **Stability of using one-epoch loss change**
>
> In SIEVE, the one-epoch loss change ΔL is used only to rank samples; thus, its practical stability is reflected directly in the stability of the selected validation set. Using the three random seeds already reported in our Waterbirds experiments, the resulting validation sets show >0.8 Jaccard similarity and ~90% overlap (table below), indicating that ΔL produces a consistent ranking and is not driven by mini-batch ordering randomness.
> | Seed pair            | (1, 2) | (1, 3) | (2, 3) |
> |-------------------|--------|--------|--------|
> | Jaccard Similarity| 0.83   | 0.80   | 0.84   |
> | Overlap (%)       | 90.49  | 89.00  | 91.09  |
>
> We have added the above results to the revised manuscript in Appendix F.
>
> ---
>
>
> ### Response to W4:
>
> **On the performance differences noted in Table 1**
>
> SIEVE is a plug-in module that replaces group-labeled validation sets, enabling existing methods to operate in a fully group-label-free setting. In Table 1, the non-SIEVE baselines (e.g., JTT, AFR, DaC) use validation set with ground-truth group labels for hyperparameter tuning and model selection, whereas the SIEVE variants do not. Consequently, the reviewer’s observation that the SIEVE versions may match or occasionally fall slightly below these baselines is expected, because the SIEVE variants have strictly less access to group-label information.
>
> Since SIEVE does not modify the training algorithms themselves, the relevant comparison is between each method using its original group-labeled validation set and the same method using a SIEVE-constructed, group-label-free validation set. The purpose of Table 1 is therefore to show that these methods maintain the worst-group performance of their original, group-labeled counterparts while using no group labels for validation. In fact, the SIEVE variants typically match the performance of the corresponding supervised versions.

---

> ### Author Response · Authors · 2025-11-30
> **Response to Reviewer YafZ (Part 2/2)**
>
> ### Response to Q1:
> **Clarification about Figure 2**
>
> In Figure 2, the non-confusing and confusing groups consists of examples with their distance to nearest other-class examples (the x-axis) larger than a high threshold and lower than a lower threshold, respectively. The sharp boundaries correspond to these two thresholds. This is consistent with our definitions of confusing and non-confusing examples in Eqs. (2) and (3). We have revised the caption of Figure 2 to clarify this.
>
> ---
>
>
> ### Response to Q2:
>
>
> **Clarification about Figure 3**
>
> Non-confusing but spurious examples are included in the “Other Examples’’ (grey) region. Figure 3 highlights the three subsets needed for illustrating the loss-dynamics intuition, and the remaining category is grouped under “Other Examples’’ for visual simplicity.
>
> ---
>
> ### Response to Q3:
>
>
> **Clarification about Lines 185-187**
>
> Thank you for the question. The statement in Lines 185–187 is intended to express that non-confusing examples are relatively easy to classify, and therefore it is possible ("may classify") for both the true and spurious decision boundaries to assign the correct label in some cases. This does not imply that all non-confusing non-spurious examples will be correctly classified by the spurious boundary.
>
> The subsequent sentence in the paper makes this scope more explicit: “the spurious decision boundary may classify some non-spurious examples correctly”, which indicates that the behavior holds only for some, not all, such examples, as illustrated in Figure 3. We have revised the wording in the paper to make this clearer.

---

### Official Review · Reviewer_Vy6F · 2025-10-30

**Soundness:** 3
**Presentation:** 2
**Contribution:** 3
**Rating:** 6
**Confidence:** 4

**Summary:**

This paper introduces SIEVE, a module that constructs a group-labeled validation set to train models robust to spurious correlations, eliminating the need for manual group annotations. The method first identifies 'confusing' training examples based on feature-space similarity and then leverages their distinct training loss dynamics: it assigns 'spurious' pseudo-labels to examples whose loss decreases rapidly and 'non-spurious' pseudo-labels to those whose loss decreases slowly or even increases. This resulting surrogate validation set can be integrated into existing robust training pipelines for model selection.

**Strengths:**

The core idea of using training dynamics to identify minority group examples is not entirely new; related concepts like learning speed have been explored in works like SSL (Zhu et al., 2023). However, the paper's primary original contribution is the two-stage filtering process. It first isolates a small, informative subset of "confusing" examples using feature-space distance, and then applies the loss dynamics analysis. The insight that the signal for separating spurious from non-spurious examples is strongest near the decision boundary, is a novel refinement that makes the approach more reliable than applying it to the entire dataset.

The empirical evaluation is of high quality and is a major strength. The authors validate their method across four standard and diverse benchmark datasets. Crucially, they don't just show that SIEVE works with a basic ERM model; they demonstrate its "plug-and-play" capability by successfully integrating it into three different state-of-the-art methods (JTT, AFR, DaC). The results consistently show that the SIEVE-generated validation set leads to performance that is competitive with, and sometimes superior to, using a ground-truth validation set.

**Weaknesses:**

The method's success hinges on the assumption that within "confusing" examples, spurious samples have rapidly decreasing loss while non-spurious ones have slowly decreasing or increasing loss. While this is convincingly demonstrated on four benchmarks, the paper could do more to explore the boundaries of this assumption. The work could be strengthened by designing a synthetic or semi-synthetic experiment where this assumption is deliberately weakened. For example, create a setting where the causal feature is highly complex and the spurious feature is only moderately easier to learn. In such a scenario, the loss dynamics might not show such a clear separation. Characterizing this failure mode would provide a deeper understanding of the method's applicability and limitations, making the paper more complete.

The paper defines confusing examples based on the minimum distance to an example from the opposite class in a pretrained feature space. This is a reasonable proxy for being near a decision boundary, but it is presented as the sole option without much justification against alternatives. The authors could improve the paper by briefly discussing alternative definitions for the "confusing" set. For instance, methods based on model uncertainty (e.g., high prediction entropy) or energy of the logits (-logsumexp) or ensemble disagreement could also identify hard-to-classify examples.

**Questions:**

1. The core of your method relies on the observation that spurious examples have rapidly decreasing loss while non-spurious ones have slowly decreasing or increasing loss. Could you elaborate on the conditions under which this assumption might fail? For instance, in a setting where the causal features are highly complex and the spurious features are only marginally simpler to learn, would the loss dynamics still show a clear enough separation for SIEVE to effectively distinguish between the groups?

2.  The selection of pseudo-spurious and non-spurious examples is governed by a quantile threshold `τ`. While the sensitivity analysis shows robustness within a narrow range, a practitioner would still need to manually set this hyperparameter. Have you explored any data-driven heuristics to automate this selection? For example, could the tails of the loss change distribution be identified automatically using outlier detection or by fitting a mixture model, thereby making the method more self-contained?

---

> ### Author Response · Authors · 2025-11-30
> **Response to Reviewer Vy6F**
>
> We thank the reviewer for acknowledging our two-stage filtering design and our high-quality empirical evaluation across diverse benchmarks and state-of-the-art methods.
>
>
>
> ---
>
> ### Response to W1 and Q1:
> **Synthetic experiment where the loss-dynamic assumption is weakened/ elaborate when the assumption might fail**
>
> Please kindly refer to the response to W2 and Q2 of Reviewer SZ7i for a summary and **Appendix D.1.4** for details.
>
>
> ---
>
> ### Response to W2:
>
> **Alternative definitions for the "confusing" set**
>
> Thanks for this great suggestion. We implemented the prediction entropy of a model fine-tuned on the training set as an alternative score, and found that, in this setting, the loss dynamics observed on our benchmarks and synthetic datasets no longer appear: both confusing & spurious and confusing & non-spurious examples show a rapid loss increase at the beginning, while the non-spurious examples achieve higher loss later and the spurious examples exhibit a quicker and sharper loss increase early in training, so the loss separation that SIEVE exploits largely disappears (see Figure 17).
>
>
> Our confusingness score was designed to depend only on general, task-agnostic features, which are independent of any model trained on the current training set that contains spurious correlations. This is because if we rely on models trained on spurious data, then examples that are inherently confusing may no longer appear confusing in the model’s learned representation space. The experiment on the alternative confusingness score confirms that this can happen.
>
>
> We have added the new results and insights to **Appendix E**.
>
> ---
>
> ### Response to Q2:
>
> **Automate the selection of $\tau$**
>
>
> It would be interesting to have a fully automatic data-driven tail-selection heuristic.
> At the moment, this appears to introduce additional hyperparameters, and we recommend using SIEVE with $\tau = 0.2$, which seems quite robust in our experiments.

---

### Official Review · Reviewer_SZ7i · 2025-10-31

**Soundness:** 2
**Presentation:** 3
**Contribution:** 3
**Rating:** 4
**Confidence:** 3

**Summary:**

This paper proposes SIEVE (Spurious-aware Iterative Validation Example selection) as a method for constructing group-aware validation sets without requiring explicit group annotations. The insight follows how confusing training examples, which are close to the decision boundary, exhibit different loss dynamics when they are spurious (rapid loss decrease) versus non-spurious (slower or even loss decrease).  SIEVE leverages this observation to iteratively select and pseudo-label validation examples, enabling robust model selection for methods that try to address spurious correlations. Experiments on four benchmark datasets show that SIEVE achieves comparable performance with methods that use true group labels when integrated.

**Strengths:**

The paper introduces a practical and modular solution for the bottleneck of needing direct group-labeled validation for robust model selection under spurious correlations.

The authors show empirically across 4 benchmark datasets that SIEVE-validated models can match or beat methods using true group labels.

The analysis on sensitivity and robustness checks for architectures and hyperparameters is indicative of the method’s stability and overall robustness.

**Weaknesses:**

I am confused about how Algorithm 2 allows training either on the full training set or with the SIEVE-selected validation samples removed. The authors claim either setting can be selected based on the desired setup. However, if the full training set option is used, evaluation on SIEVE-selected samples coud cause contamination between train and validation. Either the paper should standardize or remove the validation examples from train, or justify the effect of including it otherwise.

While the empirical observation about loss dynamics is compelling, the paper lacks concrete analysis of why this phenomenon occurs or under what conditions it holds. The conceptual explanation in Section 2.1 is intuitive but seems informal. What happens when the assumptions are violated, for eg., when loss dynamics don’t separate as cleanly, or when multiple dominant spurious features occur?

The precision of the method drops when the data contains small/hard minority groups. This can hurt methods that depend on correctly surfacing minority samples, which is quite notable for JTT-SIEVE on Waterbirds (88% to 79%). Further discussion on potential mitigation strategies are warranted.

It would be more convincing for the claims of generalisability if experiments with SIEVE go beyond vision only tasks to also include tabular or text classification tasks where there are known spurious correlations. This would also help strengthen the justification on loss dynamics.

**Questions:**

Could you clarify which setting of validation samples (with or without exclusion from training) was used in the experiments? Additionally, justify if both settings were reported?

Can you provide more formal analysis of when and why the loss dynamics pattern holds? Are there specific properties of the spurious features or data distribution that are required?

Can you investigate more deeply why JTT-SIEVE underperforms on Waterbirds? Are there modifications to SIEVE that could better serve methods requiring accurate minority group identification?

How does SIEVE extend to cases with multiple spurious attributes or continuous spurious features? The current evaluation only considers binary spurious/non-spurious divisions.

Would it be possible to include non-vision datasets (e.g. MultiNLI) to demonstrate generalization beyond vision/ CNN based setups?

---

> ### Author Response · Authors · 2025-11-30
> **Response to Reviewer SZ7i**
>
> We thank the reviewer for acknowledging our practical, modular approach and its strong, stable empirical performance across benchmarks.
>
> ---
>
>
> ### Response to W1 and Q1:
>
> **Whether validation examples were used for training**
>
> The validation examples were not used for training. We have clarified this by removing any mention of this option in Section 2.3.
>
> ---
>
>
> ### Response to W2 and Q2:
>
> **concrete analysis of when and why the loss dynamics occur/ what happens when the assumption violated**
>
> We have added a controlled synthetic experiment in **Appendix D.1.4** to systematically study when the assumption works and fails.
>
>
> Concretely, we construct a synthetic dataset where the label is a nonlinear function of a latent variable, representing a complex causal signal. We then add spurious features whose strength we control by changing (i) the proportion of examples where the spurious signal aligns with the label and (ii) the amount of noise in the spurious feature. We train a logistic regression model on this dataset and track the early loss trajectories of confusing spurious and confusing non-spurious examples.
>
> When the spurious examples are of high proportion and low noise, we observe exactly the pattern exploited by SIEVE: within the confusing examples, spurious samples exhibit a rapid loss drop, while non-spurious samples improve much more slowly or even temporarily increase in loss. This is because the strong correlation between the spurious features and the output is often exploited by the learning algorithm when learning a classifier.
>
> As we weaken the spurious signal so that the spurious feature is only marginally easier to exploit than the complex causal features, the two loss curves within the confusing set gradually move closer together and eventually become indistinguishable (Figure 14). In this setting, the loss dynamics no longer provide a clear separation, and SIEVE cannot reliably distinguish between spurious and non-spurious examples.
>
> This synthetic study therefore directly characterizes both:
> * the setting where our assumption is most effective: strong, relatively low-noise spurious features that models can quickly exploit, which broadly matches the benchmark datasets we considered.
> * The failure case highlighted by the reviewer, where spurious features are only slightly easier to learn than the causal ones and the loss trajectories no longer separate clearly.
>
> Full details of the construction, training setup, and results are provided in Appendix D.1.4 of the revised manuscript.
>
>
>
> ---
>
>
>
> ### Response to W3 and Q3
> **Why JTT-SIEVE underperforms on Waterbirds and how SIEVE could be adapted**
>
> JTT is more sensitive to the errors in the constructed validation set, because in JTT, all the model parameters are tuned based on the validation set; in contrast, AFR and DaC are less sensitive because only the last-layer parameters are tuned. We have added a more detailed discussion in Appendix C.6.
>
>
> Regarding making SIEVE more robust for methods that are sensitive to group label errors, we plan to investigate whether modeling the group label uncertainties and incorporating such uncertainties during validation is helpful in our future work. For now, we recommend using SIEVE with a robust backbone method, which demonstrates strong results in our experiments.
>
>
> ---
>
>
> ### Response to W4 and Q5:
>
> **Extension to non-vision datasets**
>
> We have added additional results on MultiNLI as below, and we observed that the loss dynamics holds on MultiNLI. See Figure 11 and Section 3 in the revised manuscript for details.
>
>
> | Method           | Worst Group Acc | Overall Acc |
> |-------------------|--------|--------|
> | ERM | 68.15 (1.71)   | 81.30 (0.18)   |
> | ERM-SIEVE      |68.44 (2.52)  | 82.04 (0.21)  |
> | JTT     | 72.50 (1.85) |  79.10 (0.30) |
> | JTT-SIEVE | 72.16 (1.34) | 79.28 (0.06) |
> | AFR     | 73.90 (0.46) | 81.67 (0.15)  |
> | AFR-SIEVE | 73.21 (1.07) | 81.34 (0.49)  |
>
>
>
> ---
>
> ### Response to Q4:
> **Extension to multiple/continuous spurious features**
>
>
> * **Multiple spurious correlations:** We have added an experiment on a synthetic dataset with two spurious features in **Appendix D.1.5**. The loss-dynamic patterns still hold in this setting and SIEVE can be applied in the same way as in the single spurious feature case.
>
>
> * **Continuous spurious features.** In all of our experiments, the model operates on continuous features. In our experiments on real data, the binary spurious features are not provided to the model as input features, but are only conceptually present, in the sense that they correspond to semantic attributes that are used to define evaluation groups. The features used by the model are learned and continuous.

---

### Author Response · Authors · 2025-11-30
**Summary of Revisions and Clarified Contributions**

We thank the reviewers for acknowledging the strengths of our work, including the novelty of the loss-dynamics observation and the resulting SIEVE algorithm, strong empirical evaluation supporting the effectiveness and robustness of SIEVE.

We have added new results and analysis and revised the paper to address reviewers' comments. Below, we clarify our work's contributions after the revision, highlighting the new results and analysis:
* SIEVE is the first plug-and-play module that can be used to remove the need for group-labeled data for various algorithms for optimizing worst-group performance for spuriously correlated data, without any group label.
* Strong empirical results demonstrating SIEVE's effectiveness and robustness. We have added results on the MultiNLI dataset to further support SIEVE's effectiveness (Section 3), and empirical analysis to demonstrate stability of SIEVE to the data ordering (Appendix F).
* A novel observation on the loss dynamics. We have added a new simulation study in **Appendix D.1.4** to provide a more formal illustration and analysis of the conceptual ideas in Section 2.1, which has been revised with reference to the new result to improve clarity:
    * The simulation study shows that in general, when both causal and spurious features are present, the loss dynamics pattern described in the Introduction arises.
    * This is because the strong correlation between the spurious features and the output is often exploited by the learning algorithm when learning a classifier.

* Extension to multiple spurious features. We have added new simulation experiments in **Appendix D.1.5** to show that the loss dynamics we observe are not limited to the single-spurious-feature case but also hold when multiple spurious features are present, and that SIEVE can be applied without modification in such settings.

For ease of review, all revisions are highlighted in blue in the updated manuscript.

---

### Meta-Review · Area_Chair_mXz2 · 2026-01-07

**Summary:**

The paper proposes a method for improving robustness to spurious corrections, in particular, to perform model selection on a validation set without access to group labels. The authors' rebuttal addressed reviewers' concerns about the stability of the method and comparisons to baselines.

**Reviewer Concerns:**

See below.

**Reviewer Scores:**

Reviewer YafZ raised questions about empirical performance gains of the proposed method, stability of the method while relying on a single epoch to determine pseudo labels and clarifying questions about the method. The authors’ rebuttal addressed these questions adequately. I especially agreed with the concern regarding the practical performance compared to baselines, but then the authors clarified the baselines have access to group labels in validation data. However, on top of original reviewer’s concerns about these performance comparisons, I believe that the authors should add a baseline of the original methods (ERM, AFR, etc) where the hyper-parameters choices / model selection happens based on the average validation accuracy (i.e. same setup as SIEVE) and compare those results. For example, AFR was shown to be robust even when model selection is done based on avg validation accuracy. Overall, I believe this reviewer would raise the score from 2 to 4.

Reviewer SZ7i raised questions about train/validation split in the proposed algorithm, stability of the loss dynamics observation, poor performance with JTT and experiments beyond vision domain. The authors responded to these questions in their rebuttal and I believe the reviewer would retain score 4 or raise the score 4->6.

Reviewer Vy6F appreciated the method’s effectiveness and practical performance and raised questions about additional synthetic “understanding” experiments and hyperparameter choices. I believe the authors addressed most questions adequately, and the reviewer would retain the score 6.

I believe the authors should demonstrate the improvements of this method on top of baselines with the same setup as SIEVE to provide strong support for their method, thus I recommend a reject.

---

### Decision · Program_Chairs · 2026-01-26

Reject